# Predicting Gait Parameters of Leg Movement with sEMG and Accelerometer Using CatBoost Machine Learning

**Alok Kumar Sharma** [1], **Shing-Hong Liu** [1,*], **Xin Zhu** [2] **and Wenxi Chen** [2]

1 Department of Computer Science and Information Engineering, Chaoyang University of Technology, 168, Jifeng E. Rd., Wufeng District, Taichung City 413310, Taiwan; rbaloksharma@gmail.com

2 Division of Information Systems, School of Computer Science and Engineering, University of Aizu, Aizu-Wakamatsu City 965-8580, Fukushima, Japan; zhuxin@u-aizu.ac.jp (X.Z.); wenxi@u-aizu.ac.jp (W.C.)

* Correspondence: shliu@cyut.edu.tw; Tel.: +86-4-2332-3000-7811

**Abstract:** This study aims to evaluate leg movement by integrating gait analysis with surface electromyography (sEMG) and accelerometer (ACC) data from the lower limbs. We employed a wireless, self-made, and multi-channel measurement system in combination with commercial GaitUp Physilog® 5 shoe-worn inertial sensors to record the walking patterns and muscle activations of 17 participants. This approach generated a comprehensive dataset comprising 1452 samples. To accurately predict gait parameters, a machine learning model was developed using features extracted from the sEMG signals of thigh and calf muscles, and ACCs from both legs. The study utilized evaluation metrics including accuracy ($R^2$), Pearson correlation coefficient (PCC), root mean squared error (RMSE), mean absolute percentage error (MAPE), mean squared error (MSE), and mean absolute error (MAE) to evaluate the performance of the proposed model. The results highlighted the superiority of the CatBoost model over alternatives like XGBoost and Decision Trees. The CatBoost's average PCCs for 17 temporospatial gait parameters of the left and right legs are $0.878 \pm 0.169$ and $0.921 \pm 0.047$, respectively, with MSE of 7.65, RMSE of 1.48, MAE of 1.00, MAPE of 0.03, and Accuracy ($R^2$-Score) of 0.91. This research marks a significant advancement by providing a more comprehensive method for detecting and analyzing gait statuses.

**Keywords:** gait parameter; surface electromyography; machine learning; CatBoost; XGBoost; decision tree

## 1. Introduction

The gait, defined as the distinctive movement pattern of the lower extremities during ambulation, serves as a key indicator of the human body's locomotive attributes [1]. This biomechanical phenomenon is integral to the control mechanisms of prosthetic legs for lower limb amputees. Additionally, gait analysis plays a critical part in the realms of individual identification, fall risk assessment, and the diagnostic process for various disorders, including Parkinson's disease [2,3]. Traditional approaches to leg movement analysis, predominantly reliant on observational techniques and complex motion capture systems, provide substantial insights but are often constrained by practical, cost, and ecological validity factors.

Surface electromyography (sEMG) is instrumental in capturing the electrical activity of muscles during leg movement, providing invaluable data on muscle coordination and activation patterns [3]. sEMGs also play a crucial role in exploring the detailed interplay between the neuromuscular system and physical movement [4]. They yield vital insights into the patterns of muscle activation, the robustness of muscular strength, and the dynamics of muscle exhaustion [5]. sEMG, an instrument essential for capturing the electrical fields emitted by muscle fibers upon contraction, utilizes motor unit action potentials (MUAPs) as its primary signal. These MUAPs, which represent the electrical patterns of muscle fibers, may increase in intensity or frequency based on the muscle's activity level. The

sEMG methodology includes a system that amplifies and visually represents these electrical signals, thereby enabling researchers and clinicians to monitor muscle activation, timing, and coordination [6]. This capability is critically important for assessing muscle functions, diagnosing neuromuscular disorders such as Myopathy, managing strength training, providing biomechanical insights, assisting in the design of orthotics and prosthetics, and advancing the development of ergonomic solutions [7]. sEMG in gait analysis proves beneficial for patients with neuromuscular conditions such as cerebral palsy, Parkinson's disease, and muscle dystrophy [8]. Several researchers have employed accelerometers (ACC) for the identification of gait parameters [9,10].

Recent advancements in gait analysis have focused on integrating inertial measurement unit (IMU) and sEMG data to enhance the estimation of gait parameters. These methods employ both sensors to capture comprehensive biomechanical movements, combining the kinematic data from IMUs with the kinetic data from sEMG sensors for a holistic view of gait dynamics. Notably, machine learning (ML) models, namely, Random Forests (RF), Neural Networks (NNs), and Support Vector Machines (SVM), are increasingly utilized to analyze the rich and multidimensional datasets measured by these sensors. This approach not only improves the accuracy of gait parameter estimations but also facilitates real-time gait analysis, making it invaluable in clinical and sports settings for immediate feedback and continuous monitoring [11,12].

The use of ML methods facilitates adaptive learning from extensive datasets, which can enhance the accuracy and effectiveness of diagnostics and monitoring in clinical environments [13]. ML has demonstrated its efficiency in analyzing sEMG signals for various applications, including gesture classification [14], muscle fatigue detection [15,16], and identifying sarcopenia [2]. In the previous research, Kidziński et al. [17] employed a data-driven method to forecast the timings of foot-contact and foot-off events by analyzing marker time series and kinematics in children exhibiting both normal and abnormal walking patterns. Arunganesh et al. [18] applied a tree-based ML model to identify lower limb movements using sEMG measurements. In another study, Rastegari et al. [19] and Trabassi et al. [20] used ML models to identify Parkinson's disease through gait analysis based on ACC data. Howcroft et al. [21] utilized different ML techniques to estimate the risk of falls using ACC parameters. Zhang et al. [22] used Support Vector Regression (SVR) models to accurately estimate essential gait phases. Jani et al. [23] employed ML techniques such as CatBoost, Random Forest (RF), XGBoost, LightGBM, and Decision Tree (DT) for detecting gait abnormalities. The result of this study indicated that CatBoost achieved the highest accuracy compared to the other ML techniques. Wu et al. [24] used traditional ML such as XGBoost, RF, and Linear Discriminant Analysis (LDA) for the categorization of lower limb movements. Among these, XGBoost performed the best. Armand et al. [25] used fuzzy decision trees to link clinical data with kinematic gait patterns of toe-walking. In the existing literature, only Liu et al. [26] have predicted 11 temporospatial gait parameters, whereas other studies [22,27] have typically predicted 2–6 gait parameters.

This study focuses on examining the temporospatial parameters of walking patterns using sEMG and ACC data of lower limbs, analyzed through ML techniques such as CatBoost, XGBoost, and DT. This study selected CatBoost, XGBoost, and DT due to their strong performance in handling complex datasets, as in similar studies [23–25]. CatBoost efficiently manages categorical data, while XGBoost is valued for its anti-overfitting capabilities. DT was chosen for its clear interpretability, which is essential in clinical applications. Together, these classifiers offer a balanced approach to our analysis. The sEMG data were collected from two primary muscles in each foot, the vastus lateralis and gastrocnemius muscles. ACCs were placed at the thighs. For benchmarking, the study utilized GaitUp Physilog® 5 shoe-worn inertial sensors, a professional gait analysis tool [28]. A wireless self-made multi-channel measurement system was employed to record muscle activity and thigh motion during treadmill running. The main contributions of this study are summarized as follows.

- Analyzed ACC and sEMG signals to extract features relevant to gait analysis, enhancing the understanding of gait dynamics.
- Applied feature selection methods to recognize the important features that contribute to the accuracy of the gait parameter predictions.
- Employed ML techniques to predict temporospatial 17 gait parameters.

Following this introduction, the study organizes the paper into several key sections. The next section details the materials and methods employed in our study. The results and discussion sections follow, providing an in-depth interpretation of the findings. The study concludes with a summary of these findings.

## 2. Materials and Methods

Figure 1 displays the architecture of our methodology, which is specifically designed for sEMG and ACC data collection by the self-made slave (right foot) and master (left foot) boards. The slave board uses XBEE S2C module to transmit data, two sEMG channels and two ACC channels, to the master board. The master board uses an HC-05 Bluetooth module to transmit data, four EMG channels, and four ACC channels, to a personal computer (PC). Additionally, GaitUp Physilog® 5 inertial sensors are attached to shoes to measure movements as references, simultaneously. Six specific parameters are derived from both sEMG and ACC data as features. Subsequently, a feature selection process is applied to pinpoint crucial features. Gait parameters are then determined using ML models. Models demonstrating superior performance underwent optimization for further enhancement. Finally, the results were scrutinized using statistical evaluation techniques.

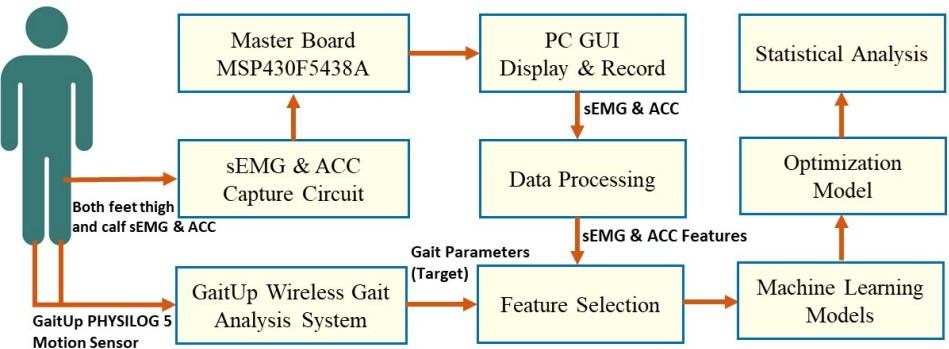

**Figure 1.** System architecture.

### 2.1. sEMG and Accelerometer Measuring Device

This study developed a self-made multi-channel wireless sEMG and ACC measurement system, as illustrated in Figure 2. The system comprises two boards, namely, a slave and a master, which utilize XBEE S2C modules for data transfer between two boards. They are placed at the left and right legs to measure the sEMGs of the thigh and calf muscles and measure the thigh motion by the x and y axes of the ACC. The sampling rate on the slave board is 1000 Hz, and the transfer rate is also 1000 Hz. The master board has 500 Hz of sampling, which includes an HC-05 Bluetooth module to send sEMG and ACC data from the slave and master boards to the PC at a frequency of 500 Hz. Dual-channel sEMG and dual-channel ACC signals equip one board. The sEMG circuitry was designed following Liu et al.'s study [16]. The system operates on a Texas Instruments (TI) MSP430F5438A microcontroller with a 12-bit analog-to-digital (ADC) resolution. Data packets to the PC have 2 header bytes and 20 data bytes. To prevent data loss from the slave board to the PC terminal, we employed a down-sampling method. The slave board transmitted data to the master board at 1000 Hz, and the master board then transmitted data to the PC terminal at 500 Hz. We found that this method effectively prevented any data loss from the slave board.

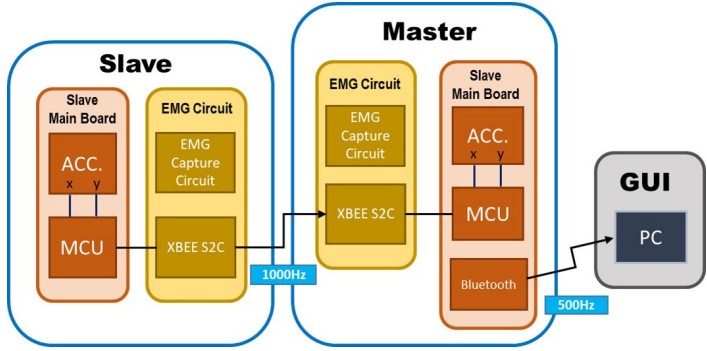

**Figure 2.** The block plot of the self-made multi-channel wireless sEMG and ACC measurement system including the slave and master boards. The slave board samples at 1000 Hz and the master board at 500 Hz.

Figure 3 displays the temporal progression of the four sEMG signals, and x and y axes of two ACCs recorded using the designed device during walking, covering a period of 6 s. The activities of the thigh and calf muscles of the left and right legs show the time sequences. The x and y axes of the accelerometers (ACCs) on the left and right legs exhibit a similar phenomenon, where the x-axis signal is significantly oriented toward the calf muscle activity. Thus, we hypothesize that the gait status could be described by these signals.

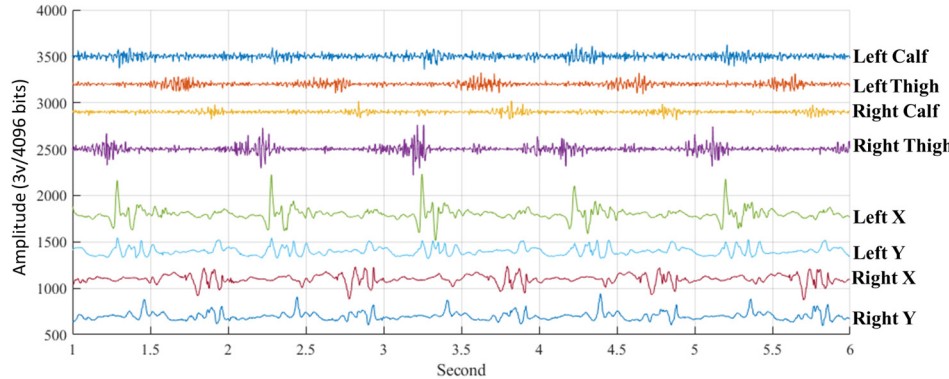

**Figure 3.** sEMGs and ACC signals under walking.

### 2.2. Experiment Protocol

The study involved voluntary participants who were adults diagnosed with sarcopenia, had healthy limbs, and maintained normal standing positions. The group included 17 individuals aged 19 to 23, with an average age of $20 \pm 1$ years. The average height of the individuals was $156 \pm 4.6$ cm, with a range of 149 to 164 cm. Similarly, the average weight was $45.9 \pm 5.7$ kg, with a range of 31 to 56 kg. The average shoe size among participants was $23.9 \pm 0.6$, varying from 23 to 25 cm. The four electrodes were placed on the gastrocnemius and vastus lateralis muscles of the left and right legs for sEMG measurements, the ACC was placed on the outer thigh for recoding the thigh motion, and the GaitUp Physilog® [28–30] shoe-worn inertial sensors were positioned on the tongues of shoes for gait status measurement. Participants evaluated their physical health before participating in the experiment. The Chung-Shan University Hospital's Research Ethics Committee in Taichung City, Taiwan, approved this experiment with the reference number CS2-22210.

### 2.3. Data Processing

The study recorded data for 6 min, totaling 180,000 data points for both sEMGs and ACCs. Data segmentation started with intentional muscle contractions in sEMG recordings and activity onset in the GaitUp Lab, marking the beginning of data slicing. This step is critical as it marks the point where significant gait events begin to be recorded, ensuring

that the analysis focuses on relevant data. To ensure the integrity of our analysis and mitigate the impact of potential outliers or anomalies at the start and end of each gait cycle, we strategically omitted 7.5 s from both the start and the end of a signal. The data, including two-channel sEMGs, x- and y-axis signals of ACC and GaitUp Physilog® 5 sensor readings, were divided into 30 s batches with a 15 s window shift, resulting in 22 sample sets per experiment. Gait parameters were derived by segmenting and processing the data captured by GaitUp Physilog® 5 using the GaitUp Lab software (Gait Up SA, CH-1015 Renens, Switzerland), which was the target output.

### 2.3.1. Gait Parameters

The GaitUp Lab gait analysis system assessed 17 left and right foot gait parameters, including 9 temporal and 8 spatial parameters. Temporal parameters include Heel Strike (HS), Gait Cycle Time (gct), Double Leg Support (DS), Cadence, Stance, and Swing Ratios, Foot Flat Ratio (FFr), Push Ratio (Purify), and Load Ratio (LDr). These parameters measure aspects such as the heel's contact with the ground, the duration and phases of the gait cycle, and weight distribution.

Spatial parameters include Stride Length, Step Length, Gait Speed, Peak Swing (maximum leg angular velocity during the swing phase), Foot Pitch Angles at Heel Strike (HSP) and Toe-Off (TOP), Swing Width, and 3D Path Length. These focus on the distance, speed, and angles involved in walking movements. This categorization offers a detailed view of the dynamics of human gait. The gait parameters are described in Table 1.

**Table 1.** Description of 17 temporospatial gait parameters.

| Type | Name | Units | Description |
|---|---|---|---|
| Temporal Parameter | Heel Strike (HS) | Seconds | The moment the heel makes contact with the ground. |
| | Gait Cycle Time (GCT) | Steps/minute | Gait cycle duration is the time between heel strikes on the same foot. |
| | Double Leg Support (DS) | % of cycle duration | The bipedal stance period is when both feet are on the ground during the gait cycle. |
| | Cadence | Steps/minute | The number of steps walked per minute. |
| | Stance | % of cycle duration | The foot hits the ground in the gait cycle's stance phase. |
| | Swing | % of gait cycle | The swing phase refers to the interval when the foot is not in contact with the ground during the gait cycle. |
| | Foot Flat Ratio (FFr) | % of stance | The phase of the stance in which the foot is completely in contact with the ground, with the sole entirely touching the surface. |
| | Push Ratio (Purify) | % of stance | The time between flat soles and lifted toes in stance. |
| | Load Ratio (LDr) | % of stance | The stance period from heel strike to full sole contact. |
| Spatial Parameters | Step Length | meters | The spatial distance between the feet when positioned on the ground. |
| | Stride Length | meters | Distance between heel strikes, equaling one gait cycle. |
| | Gait Speed | m/s | Speed of forward walking. |
| | Peak Swing | m/s | Maximum angular velocity from heel to toe during swing. |
| | Foot Pitch Angle at Heel Strike (HSP) | degree | The angle formed by the foot's contact with the ground upon impact. |
| | Foot Pitch Angle at Toe-Off (TOP) | degree | The angle of the toes relative to the ground just before lift-off at the end of the propulsion phase. |
| | Swing Width | meters | The largest sideways distance in the swing phase corresponds to the maximum lateral offset. |
| | 3D Path Length | % of stride length | Depicts the scaled trajectory of the 3D gait cycle using stride length. |

### 2.3.2. Signal Parameters

We used six frequency- or time-domain parameters derived from the sEMGs and ACC signals. The Mean Frequency (MF) is calculated by multiplying the signal's power

spectrum by its frequency and then separating this product by the power spectrum sum. Equation (1) defines the MF formula.

$$\text{MF} = \sum_{j=1}^{D} f_j p_j \Big/ \sum_{j=1}^{D} p_j, \tag{1}$$

where $f_j$ denotes the $j$th frequency in the signal corresponding to the $j$th spectrum density, $p_j$, and $D$ is the number of the discrete Fourier transform. The median frequency (MDF) divides the power spectrum of the signal into two equal parts, as depicted in Equation (2).

$$\sum_{j=0}^{MDF} p_j = \sum_{j=MDF}^{D} p_j = \frac{1}{2} \sum_{j=0}^{D} p_j, \tag{2}$$

Equation (3) presents the standard deviation (STD).

$$\text{STD} = \sqrt{\frac{1}{2} \sum_{i=1}^{N} (x_i - \overline{x})^2}, \tag{3}$$

where $x_i$ represents the sampled data, $\overline{x}$ represents mean of $x$, and $N$ denotes the size of the window. Equation (4) represents the root mean square (*RMS*).

$$\text{RMS} = \sqrt{\frac{\sum_{i=1}^{N} x_i^2}{N}}. \tag{4}$$

Sample Entropy (SampEn) is used to calculate the intricacy and regularity of a signal's temporal sequence. An increased entropy value indicates time series complexity. When the length of a signal is $S$, calculating sample entropy requires defining dimension $m$ and selecting a suitable $v$ value. After determining $m$, a segment signal was labeled $X_m$. The method is outlined as stated below,

$$X_m(i) = \{x_i, x_{i+1}, x_{i+2}, \ldots, x_{i+m-1}\}. \tag{5}$$

Equation (5) is the formula for SampEn.

$$\text{SampEn} = -ln\frac{Y}{Z}, \tag{6}$$

where

$$Y = d[X_{m+1}(i), X_{m+1}(j)] < v, \tag{7}$$

$$Z = d[X_m(i), X_m(j)] < v. \tag{8}$$

The function $d[X_m(i), X_m(j)]$ measures the distance between vectors $X_m(i)$ and $X_m(j)$ in the reconstructed phase space, where each vector represents a subsequence of length starting at points $i$ and $j$, respectively. The limits for $i$ and $j$ in Equations (7) and (8) are intended to span the entire dimension of the time series, ensuring that all possible pairs of subsequences of length $m$ and $m + 1$ are considered in computation. Typically, $i$ and $j$ range from 1 to $S$-$m$ or $S$-$m$-1. This research sets $m = 2$ and $v = STD \times 0.2$. The variables $m$ and $v$ in our SampEn calculations are chosen based on our best adjustment, and following the methodology of previous studies [31,32].

In our study, these parameters were carefully chosen for their relevance to signal characterization. MF and MDF are critical for understanding the spectral attributes and detecting muscle fatigue [33]. STD evaluates signal variability, offering insights into motor control [34]. SampEn measures data complexity, indicating physiological conditions' predictability [31]. Lastly, RMS is utilized to assess the overall signal magnitude, which correlates with muscle activation levels [35].

### 2.4. Feature Selection

In this research, the XGBoost model was used to determine key features for predicting leg movements in gait parameters. XGBoost assesses feature importance by calculating scores to reflect each feature's utility in the model's decision trees. Features are evaluated based on frequency (how often they appear in trees), gain (impact on model accuracy), and coverage (number of data points a feature affects). These metrics help pinpoint the most influential features, aiding in model refinement and feature selection. Initially, the study extracted 84 features from the ACC and sEMG signals. Subsequently, a feature selection method was applied to identify the useful features for leg movement, resulting in the identification of 8 key features. Table 2 displays these eight important features used for predicting gait parameters.

**Table 2.** Identified important features.

| No. | Features | Type |
|-----|----------|------|
| 1 | MF_X-axis | ACC |
| 2 | MDF_X-axis | ACC |
| 3 | STD_X-axis | ACC |
| 4 | STD_Y-axis | ACC |
| 5 | RMS_X-axis | ACC |
| 6 | RMS_Y-axis | ACC |
| 7 | SampEn of Calf | sEMG |
| 8 | SampEn of Thigh | sEMG |

### 2.5. Machine Learning Models

Three widely recognized regression models, CatBoost, XGBoost, and DT, were used by Python. This research involved training regression models on a system equipped with an Intel i7-10700 CPU and an NVIDIA GeForce RTX3070 graphics card, and 64 GB of 2933 MHz RAM. The following subsections offer a concise summary of each model used in the study.

#### 2.5.1. CatBoost

CatBoost [36] is an advanced open-source ML algorithm developed by Yandex, renowned for its efficient handling of categorical data and high performance. It stands out in the gradient-boosting landscape due to its native processing of categorical variables, eliminating the need for extensive preprocessing. CatBoost [37] utilizes a combination of categorical features, leveraging the relationships among these features to significantly enhance the dimensionality of the feature space. To minimize overfitting and enhance both the accuracy and generalizability of the model, CatBoost employs a perfectly symmetrical tree structure. Known for its speed and accuracy, CatBoost delivers robust results even with default settings, making it a favored choice among data scientists and researchers for a wide range of applications.

#### 2.5.2. XGBoost

XGBoost, or eXtreme Gradient Boosting [38], represents an enhancement over traditional Gradient Boosting methods. Structurally similar to DT, it combines several weak DTs to create a powerful predictive tool. XGBoost generally outperforms standard classification and regression techniques in accuracy. Due to its robust adaptive learning capabilities, XGBoost remains a favored choice for both regression and classification tasks in current studies and competitive environments. XGBoost [39] addresses the issue of overfitting by regulating tree complexity and incorporating a regularization term into an objective function, enhancing model reliability.

### 2.5.3. Decision Tree

Decision Tree [40] is a model defined by a tree-like structure of hierarchy. Each node in this structure represents different decisions, ultimately guiding toward the expected results. The unique setup of DTs makes them exceptionally transparent, allowing for a straightforward understanding of the model's decision-making rules. Therefore, DTs [41] have gained popularity as one of the leading nonlinear regression models. It is especially skilled in handling nonlinear regression challenges, effectively capturing complex relationships between inputs and outputs without relying on linear associations. DT is capable of developing several linear regression models within the leaf nodes of the tree. Furthermore, DT visually demonstrates the significance of various contributors, thereby enhancing analytical support. Nonetheless, their dependency on categorical variables poses a significant challenge [42].

### 2.6. Statistical Analysis

This study has employed evaluation metrics, namely, Accuracy ($R^2$), Mean Squared Error (MSE), Mean Absolute Error (MAE), Root Mean Squared Error (RMSE), and Mean Absolute Percentage Error (MAPE), to assess the model's performance. MSE calculates the mean of the squared deviations between predicted and observed values.

$$\text{MSE} = \frac{1}{n}\sum_{i=1}^{n}\left(Yi - \hat{Y}i\right)^2. \tag{9}$$

Here, $n$ represents the total number of data points, $Y_i$ refers to the actual value, and $\hat{Y}_i$ indicates the predicted value.

RMSE is a measure of the square root of the mean of the squares of the errors. It is essentially the square root of MSE.

$$\text{RMSE} = \sqrt{\frac{1}{n}\sum_{i=1}^{n}\left(Yi - \hat{Y}i\right)^2}. \tag{10}$$

MAE calculates the average size of errors in a set of predictions, disregarding the direction of the errors.

$$\text{MAE} = \frac{1}{n}\sum_{i=1}^{n}\left|Yi - \hat{Y}i\right|. \tag{11}$$

MAPE measures the average of the absolute percentage errors by comparing the prediction to the actual value.

$$\text{MAPE} = \frac{100}{n}\sum_{i=1}^{n}\left|\frac{Yi - \hat{Y}i}{Yi}\right|. \tag{12}$$

The $R^2$ score, also known as the coefficient of determination, quantifies how much the variance in the dependent variable is explained by the independent variables in a regression model. It is a crucial indicator of model fit, with a score of 1 signifying perfect prediction accuracy.

$$R^2 = 1 - \frac{\text{SSR}}{TSS} \tag{13}$$

where

- SSR (Sum of Squares of Residuals) calculates the sum of the squared differences between the observed values and the values predicted by the model. It is mathematically represented as:

  $\sum_{i=1}^{n}\left(Y_i - \hat{Y}_i\right)^2$, where $Y_i$ are the actual values and $\hat{Y}_i$ denotes the prediction values.

- TSS (Total Sum of Squares) represents the overall variance within the dataset, determined by summing the squared deviations of each observed value from the dataset's mean:

$\sum_{i=1}^{n} (Y_i - \overline{Y})^2$, where $\overline{Y}$ represents the mean of observed values.

This study presents the numerical data as a mean (M) $\pm$ standard deviation (STD). To elucidate the connection between the target and predicted values observed in our test dataset, we have employed the Pearson correlation coefficient (PCC). The mathematical formulation of this coefficient is explicitly detailed in Equation (14),

$$\text{PCC} = \frac{\sum_{i=1}^{n} (Y_i - m_Y)(\hat{Y}_i - m_{\hat{Y}})}{\sqrt{\sum_{i=1}^{n} (Y_i - m_Y)^2 \sum (\hat{Y}_i - m_{\hat{Y}})^2}}, \tag{14}$$

where $m_Y$ and $m_{\hat{Y}}$ correspond to the mean values of the actual values, $Y_i$, and the prediction values, $\hat{Y}_i$, respectively. This statistical approach enables a rigorous evaluation of the linear correlation between the two sets of variables under investigation.

## 3. Results

The study employed CatBoost, XGBoost, and DT algorithms to predict gait parameters. Features extracted from the parameters of signals served as input variables, with the GaitUp value used as a target variable. The dataset was separated into training (80%) and testing (20%) subsets, with the testing dataset used to evaluate the trained models. Table 3 shows the PCCs for predicting 17 left and right foot gait parameters. Where CatBoost and XGBoost predict 15 gait parameters correctly with a PCC value above 0.80 for both feet. CatBoost and XGBoost demonstrated superior PCC of $0.878 \pm 0.169$ and $0.887 \pm 0.174$ for the left foot, and $0.921 \pm 0.047$ and $0.906 \pm 0.052$ for the right foot, correspondingly. Conversely, the DT model showed the least effective results, with coefficients of $0.807 \pm 0.157$ for the left foot and $0.795 \pm 0.046$ for the right foot. The study found that the CatBoost and XGBoost models demonstrate nearly equal performance. Both belong to the category of ensemble learning methods, which combine results from multiple models to enhance stability and generalization, thereby reducing the risk of overfitting while maintaining superior performance. Thus, the performances of the two models would be similar. However, the memory constraints and running time of CatBoost in the training phase are better than the XGBoost [36]. However, CatBoost is sensitive to the hyper-parameter settings. Ensemble learning methods are primarily divided into two categories: bagging and boosting. DT falls under the bagging category, which constructs models simultaneously using random subsets of data and combines the predictions from all models. Azmi et al. [43] showed that XGBoost has a better performance than DT with a large amount of unstructured data. In this study, 17 gait parameters were predicted. Thus, the boosting method should be used.

Table 4 displays a performance comparison of three different ML models: CatBoost, XGBoost, and DT, using various error metrics such as MSE, RMSE, MAE, and MAPE. CatBoost and XGBoost exhibit very similar performances, with CatBoost having a slight edge. For example, CatBoost shows a MSE of 7.65 and a RMSE of 1.49, indicating a lower average squared error, compared to XGBoost's MSE of 7.81 and RMSE of 1.53. Both models have a MAE of 1.00 and a MAPE of 0.03, suggesting strong predictive accuracy with low average and percentage errors. In contrast, the DT model underperforms, with higher error rates across all metrics: a MSE of 23.22, RMSE of 2.56, MAE of 1.49, and a MAPE of 0.04. These higher values indicate less accuracy in predicting outcomes compared to CatBoost and XGBoost. This study also calculates the accuracy ($R^2$) shown in Table 5. Figure 4 shows the results of MSE (blue), RMSE (orange), MAE (light orange), MAPE (gray), and $R^2$-Score (green) for the different models. The results show that CatBoost achieved a model accuracy ($R^2$) of 91%, outperforming XGBoost, which has an 81% accuracy ($R^2$). The DT model trails with 65% accuracy, underscoring its lower effectiveness in comparison to CatBoost and XGBoost.

**Table 3.** PCC of 17 temporospatial gait parameters for left and right feet using ACC and sEMG features.

| Name | CatBoost | | XGBoost | | DT | |
|---|---|---|---|---|---|---|
| | L | R | L | R | L | R |
| HS | 0.77 | 0.82 | 0.78 | 0.77 | 0.84 | 0.85 |
| GCT | 0.93 | 0.95 | 0.94 | 0.94 | 0.85 | 0.86 |
| Cadence | 0.91 | 0.93 | 0.92 | 0.93 | 0.83 | 0.84 |
| Stance | 0.96 | 0.94 | 0.96 | 0.91 | 0.92 | 0.76 |
| Swing | 0.96 | 0.94 | 0.95 | 0.91 | 0.92 | 0.76 |
| LDr | 0.91 | 0.93 | 0.93 | 0.93 | 0.80 | 0.72 |
| FFr | 0.94 | 0.95 | 0.95 | 0.93 | 0.79 | 0.82 |
| PUr | 0.93 | 0.95 | 0.94 | 0.92 | 0.81 | 0.79 |
| DS | 0.96 | 0.96 | 0.95 | 0.95 | 0.92 | 0.83 |
| Stride Length | 0.92 | 0.93 | 0.93 | 0.91 | 0.82 | 0.85 |
| Gait Speed | 0.80 | 0.84 | 0.80 | 0.84 | 0.70 | 0.75 |
| Peak Swing | 0.96 | 0.97 | 0.96 | 0.96 | 0.88 | 0.84 |
| HSP | 0.96 | 0.96 | 0.96 | 0.95 | 0.89 | 0.77 |
| TOP | 0.96 | 0.96 | 0.96 | 0.96 | 0.89 | 0.82 |
| Swing Width | 0.94 | 0.92 | 0.94 | 0.91 | 0.85 | 0.75 |
| 3D Path Length | 0.26 | 0.85 | 0.24 | 0.83 | 0.24 | 0.73 |
| Step Length | 0.86 | 0.86 | 0.87 | 0.86 | 0.77 | 0.78 |
| **Mean** | **0.878** | **0.921** | **0.881** | **0.906** | **0.807** | **0.795** |
| **STD** | **0.169** | **0.047** | **0.174** | **0.052** | **0.157** | **0.046** |

**Table 4.** Performance metrics of ML models using ACC and sEMG features.

| Models | MSE | RMSE | MAE | MAPE |
|---|---|---|---|---|
| CatBoost | **7.65** | **1.49** | **1.00** | **0.03** |
| XGBoost | 7.81 | 1.53 | 1.00 | 0.03 |
| DT | 23.22 | 2.56 | 1.49 | 0.04 |

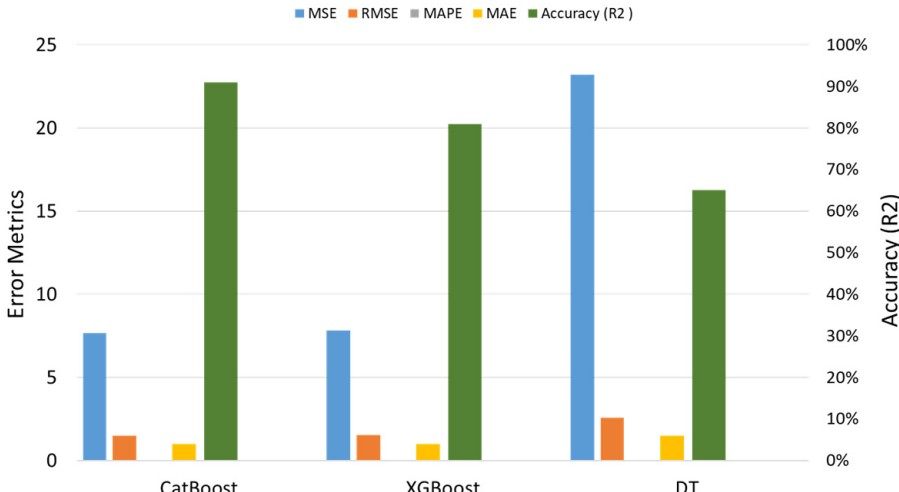

**Figure 4.** The results of MSE (blue), RMSE (orange), MAE (light orange), MAPE (gray), and $R^2$-Score (green) for CatBoost, XGBoost, and DT models.

**Table 5.** Model accuracy ($R^2$) using ACC and sEMG features.

| Models | Accuracy ($R^2$) |
|---|---|
| CatBoost | 91% |
| XGBoost | 81% |
| DT | 65% |

This study also focused on predicting gait parameters using only ACC features, as shown in Table 6. For this purpose, the CatBoost model was employed. The resulting mean ± STD values were $0.846 \pm 0.174$ for the left and $0.886 \pm 0.40$ for the right feet. Comparing these results with those obtained using a combination of ACC and sEMG features, it was evident that the combined features yielded better outcomes.

**Table 6.** PCC for 17 temporospatial gait parameters of left and right feet using CatBoost with ACC features.

| Name | L | R |
|---|---|---|
| HS | 0.69 | 0.81 |
| GCT | 0.90 | 0.92 |
| Cadence | 0.88 | 0.90 |
| Stance | 0.94 | 0.90 |
| Swing | 0.94 | 0.90 |
| LDr | 0.88 | 0.86 |
| FFr | 0.89 | 0.89 |
| PUr | 0.89 | 0.90 |
| DS | 0.93 | 0.93 |
| Stride Length | 0.89 | 0.90 |
| Gait Speed | 0.78 | 0.82 |
| Peak Swing | 0.93 | 0.93 |
| HSP | 0.92 | 0.89 |
| TOP | 0.94 | 0.94 |
| Swing Width | 0.90 | 0.92 |
| 3D Path Length | 0.22 | 0.83 |
| Step Length | 0.84 | 0.84 |
| **Mean ± STD** | **0.846 ± 0.174** | **0.886 ± 0.40** |

## 4. Discussion

The gait analysis is a finite element analysis, which is a key component of modern therapeutic and rehabilitative programs for observing patients' walking. Pheasant et al. [44] presented "Human Walking" and were the first to study gait kinematics using video camera techniques for human walking. This method captures the kinematic dynamics in the spatial and temporal domains when subjects walk [45]. The sEMG signals of subjects are also measured to explore the muscles' conditions. The limitation of the video camera technique is that subjects will be measured at a motion laboratory. Now, the IMUs, like accelerometers and gyroscopes, are used to measure the temporospatial parameters of gaits [28]. The advantage of the IMUs is that subjects can naturally walk on the ground. However, this method also needs to combine the sEMG measurement to understand the muscle activities. Muro-de-la-Herran et al. [46] reviewed publications from 2012 and 2013 related to gait analysis techniques and their clinical applications. Out of the thirty-two articles they found, 40% discussed video camera systems, 37.5% dealt with systems based on IMUs, and 22.5% examined various other wearable sensor systems. This study used sEMG and ACC to predict the 17 temporal–spatial parameters of gaits and approach to 0.91 accuracy ($R^2$). Our proposed method uses sEMG and IMUs to predict gait parameters. Out of 17 parameters, 15 were accurately predicted by the CatBoost model, in which PCCs of the left and right feet were larger than 0.800. At the same time, the sEMG signals provide valuable information about muscle activities. In this study, two gait parameters—3D Path Length and HS—

showed lower PCC values compared to others. The PCCs of Path Length for the left and right feet have a very large difference (0.24 vs. 0.83). But, in the previous study [26], PCCs of Path Length were 0.90 and 0.90 for left and right feet with XGBoost. This result showed that the features of ACC could not be used to predict the Path Length. According to Cruz-Jentoft et al. [47], sarcopenia typically leads to reductions in muscle mass and strength, which are correlated with slower walking speeds, shorter stride lengths, and increased gait variability. These results reflect the muscle weakening associated with the condition and its impact on the functionality and stability of the lower limbs during movement. Therefore, this method provides more comprehensive insights into gait analysis. These gait parameters are crucial for diagnosing gait disorders and designing assistive devices as well as the customization of prosthetics and orthotics.

This study also compares our findings with previous studies. Zhang et al. [22] used ML to estimate three gait parameters—stride length, velocity, and foot clearance—with MAEs of $2.37 \pm 0.65$, $3.10 \pm 0.57$, and $0.58 \pm 0.28$, respectively, for running. The MAE for 17 gait parameters in this study, predicted by CatBoost, is 1.00, which represents an improvement over the previous study. Moreover, Liu et al. only estimate 11 temporospatial gait parameters (six temporal parameters and five spatial parameters), with PCCs above 0.800, by sEMG signals [26]. In this study, the number of predicted gait parameters with PCCs above 0.800 was fifteen when the sEMG signals were combined with the ACC signals. In another study, Wei et al. [27] utilized sEMG signals to recognize gait phases and predict lower limb kinematics. The research employed LDA, SVM, and LSTM as predictive models, collecting sEMG data from nine leg muscles to estimate four gait parameters: initial contact, foot flat, heel-off, and toe-off. The study found that LSTM achieved the highest performance with an accuracy of 93% and a RMSE of 1.86. In comparison, our study achieved a better RMSE of 1.46 and demonstrated superior PCC values. The results show that the IMU and ACC could describe in more detail leg-motion parameters when the user walking or running.

## 5. Conclusions

This study marks a notable advancement in biomechanics, specifically in the analysis of leg movements, by integrating gait analysis with sEMG and ACC data. Utilizing advanced wireless technology, researchers gathered a dataset from 17 participants, enhancing our understanding of human locomotion. A significant aspect of this research is the use of an ML model to predict gait parameters from sEMG signals of thigh and calf muscles, along with ACC signals, demonstrating high accuracy. The CatBoost model's superiority over other models, such as XGBoost and Decision Tree, especially in processing complex high-frequency sEMG and ACC signals, is a key finding. This suggests potential for broader application in biomechanical data analysis. The study has limitations, including small datasets and challenges in improving the PCC results, as it could not accurately predict HS and Path Length. Future studies should consider enlarging the sample population and integrating neural network and deep learning approaches to achieve a more thorough and dependable analysis of gait.

**Author Contributions:** Conceptualization, S.-H.L.; methodology, A.K.S. and S.-H.L.; software, A.K.S.; validation, X.Z. and W.C.; investigation, S.-H.L.; data curation, S.-H.L., X.Z., and W.C.; writing—original draft preparation, A.K.S. and S.-H.L.; writing—review and editing, S.-H.L., X.Z., and W.C.; supervision, S.-H.L.; project administration, S.-H.L. and X.Z.; funding acquisition, S.-H.L. All authors have read and agreed to the published version of the manuscript.

**Funding:** The funding for this research was provided by the National Science and Technology Council, Taiwan, under grant NSTC 111-2221-E-324-003-MY3.

**Institutional Review Board Statement:** The research adhered to the Helsinki Declaration's principles and received ethical approval from the Research Ethics Committee at Chung-Shan University Hospital in Taichung City, Taiwan (Approval No. CS2-22210).

**Data Availability Statement:** Data are contained within the article.

**Conflicts of Interest:** The authors declare no conflicts of interest.

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
