# Peer review of "Predicting Gait Parameters of Leg Movement with sEMG and Accelerometer Using CatBoost Machine Learning"

_electronics, doi:10.3390/electronics13091791_

Round 1
Reviewer 1 Report
Comments and Suggestions for Authors
· Abstract of the manuscript needs to be revised. Authors are highlighting the values of Pearson’s Correlation Coefficient. But the values of accuracy and other performance measures should also be highlighted for most performing classifier in the abstract part.
· Introduction part is very brief. Authors should highlight the nobility of the research in this section.
· The major contributions should be highlighted in enumerated form
· Add one paragraph about section wise organization of the paper at the end of introduction section.
· Literature survey part is missing. Add brief literature survey in the Introduction section.
· Resolutions of Figure 1 and 2 need to be improved
· Discuss gait parameters mentioned in section 2.3.1. in more detail
· Why have authors used given signal parameters during feature extraction process? Specify reason
· Elaborate feature selection procedure in more detail
· Why have authors preferred CatBoost, XGBoost and Decision tree classifiers only over other classifiers?
· The accuracy in Table 4 should be mentioned in terms of percentage
· Authors should use graphical representation of results during analysis of results.
· Add a table of existing works in the same domain and compare it with the proposed work in Results and Discussion section.
· Highlight the limitations of the project work
Comments on the Quality of English LanguageProfessional English editing is required.
Author Response
To Reviewer #1:
Thank the reviewer for his/her valuable comments that make better this manuscript. The texts in this revised manuscript have been corrected/ modified by yellow mark. It is our sincere hope that this revision will enhance readability and strengthen of the manuscript to satisfy the requirements of this prestigious journal.
Comments and Suggestions for Authors
- Abstract of the manuscript needs to be revised. Authors are highlighting the values of Pearson’s Correlation Coefficient. But the values of accuracy and other performance measures should also be highlighted for most performing classifier in the abstract part.
Response: Thank reviewer for your comments and suggestion. We address this comment in abstract section. Lines 10-24.
Abstract: This study aims to evaluate leg movement by integrating gait analysis with surface electromyography (sEMG) and accelerometer (ACC) data from the lower limbs. We employed a wireless, self-made, multi-channel measurement system in combination with the commercial, GaitUp Physilog®5 shoe-worn inertial sensors to record the walking patterns and muscle activations of 17 participants. This approach generated a comprehensive dataset comprising 1,452 samples. To accurately predict gait parameters, a machine learning model was developed using features extracted from the sEMG signals of thigh and calf muscles and ACCs from both legs. The study utilized evaluation metrics including accuracy (R2), Pearson correlation coefficients (PCC), root mean squared error (RMSE), mean absolute percentage error (MAPE), mean squared error (MSE), and mean absolute error (MAE) to evaluate the model's performance. The outcomes highlighted the superiority of the Catboost model over alternatives like XGBoost and Decision Trees. The Catboost average PCC for 17 temporospatial gait parameters for the left and right legs are 0.878 ± 0.169 and 0.921 ± 0.047, respectively, with MSE of 7.65, RMSE of 1.48, MAE of 1.00, MAPE of 0.03, and Accuracy(R2) of 0.91. This research marks a significant advancement in the field by providing a more comprehensive method for detecting and analyzing gait statuses.
- Introduction part is very brief. Authors should highlight the nobility of the research in this section.
Response: Thank reviewer for your comments and suggestion. We extend the Introduction section of our paper. Lines 28 -107.
- Introduction
The gait, defined as the distinctive movement pattern of the lower extremities during ambulation, serves as a key indicator of the human body's locomotive attributes [1]. This biomechanical phenomenon is integral to the control mechanisms of prosthetic legs for lower limb amputees. Additionally, gait analysis plays a critical part in the realms of individual identification, fall risk assessment, and the diagnostic process for various disorders, including Parkinson's disease [2][3]. Traditional approaches to leg movement analysis, predominantly reliant on observational techniques and complex motion capture systems, provide substantial insights but are often constrained by practical, cost, and ecological validity factors.
Surface electromyography (sEMG) is instrumental in capturing the electrical activity of muscles during leg movement, providing invaluable data on muscle coordination and activation patterns [3]. In exploring the detailed interplay between neuromuscular system and physical movement, sEMGs play a crucial role [4]. They yield vital insights into the patterns of muscle activation, the robustness of muscular strength, and the dynamics of muscle exhaustion [5]. sEMG, an instrument essential for capturing the electrical fields emitted by muscle fibers upon contraction, utilizes motor unit action potentials (MUAPs) as its primary signal. These MUAPs, which represent the electrical patterns of muscle fibers, may increase in intensity or frequency based on the muscle's activity level. The sEMG methodology includes a system that amplifies and visually represents these electrical signals, thereby enabling researchers and clinicians to monitor muscle activation, timing, and coordination [6]. This capability is critically important for assessing muscle function, diagnosing neuromuscular disorders such as Myopathy, managing strength training, providing biomechanical insights, assisting in the design of orthotics and prosthetics, and advancing the development of ergonomic solutions [7]. sEMG in gait analysis proves beneficial for patients with neuromuscular conditions such as cerebral palsy, Parkinson's disease and muscle dystrophy [8]. Several researches have employed accelerometer (ACC) for the identification of gait parameters [9,10].
Recent advancements in gait analysis have focused on integrating inertial measurement unit (IMU) and sEMG data to enhance the estimation of gait parameters. These methods employ both types of sensors to capture comprehensive biomechanical movements, combining the kinematic data from IMUs with the kinetic data from sEMG sensors for a holistic view of gait dynamics. Notably, machine learning (ML) models namely, Random Forests (RF), Neural Networks (NNs), and Support Vector Machines (SVM) are increasingly utilized to analyze the rich, multidimensional data sets provided by these sensors. This approach not only improves the accuracy of gait parameter estimations but also facilitates real-time gait analysis, making it invaluable in clinical and sports settings for immediate feedback and continuous monitoring [11,12].
The use of ML methods facilitates adaptive learning from extensive datasets, which can enhance the accuracy and effectiveness of diagnostics and monitoring in clinical environments [13]. ML has demonstrated its efficiency in analyzing sEMG signals for various applications, including gesture classification [14], muscle fatigue detection [15][16], and identify sarcopenia [2]. In the previous research Kidziński et al. [17] employed a data-driven method to forecast the timings of foot-contact and foot-off events by analyzing marker time series and kinematic in children exhibiting both normal and abnormal walking patterns. Arunganesh et al. [18] applied a tree-based ML model to identify lower limb movements using sEMG measurements. In another study, Rastegari et al. [19] and Trabassi et al. [20] used ML models to identify Parkinson's disease through gait analysis based on ACC data. Howcroft et al. [21] utilized different ML techniques to estimate the risk of falls using ACC parameters. Zhang et al. [22] used Support Vector Regression (SVR) models to accurately estimate essential gait phases. Jani et al. [23] employed ML techniques such as CatBoost, Random Forest (RF), XGBoost, LightGBM, and Decision Tree (DT) for detecting gait abnormalities. The outcome of this study indicated that CatBoost achieved the highest accuracy compared to the other ML techniques. Wu et al. [24] used traditional ML such as XGBoost, RF, and Linear Discriminant Analysis (LDA) for the categorization of lower limb movements. Among these, XGBoost performed the best. Among these, XGBoost performed well. Armand et al. [25] used fuzzy decision trees to link clinical data with kinematic gait patterns of toe-walking. In the existing literature, only Liu et al. [26] have predicted 11 temporospatial gait parameters, whereas other studies [27] [22] have typically predicted between 2 and 6 gait parameters.
This study focuses on examining the temporal and spatial parameters of walking patterns using sEMG and ACC data of lower limbs, analyzed through ML techniques such as Catboost, XGBoost and DT. This study selected CatBoost, XGBoost, and DT due to their strong performance in handling complex datasets, as similar studies [23][24][25]. CatBoost efficiently manages categorical data, while XGBoost is valued for its anti-overfitting capabilities. DT were chosen for their clear interpretability, which is essential in clinical applications. Together, these classifiers offer a balanced approach to our analysis. The sEMG data were collected from two primary muscles in each foot, the vastus lateralis and gastrocnemius muscles. ACC were placed at the thighs. For benchmarking, the study utilized the GaitUp Physilog®5 shoe-worn inertial sensors, a professional gait analysis tool [28]. A wireless self-made multi-channel measurement system was employed to record muscle activity and thigh motion during treadmill running. The main contribution of this study:
- Analyzed ACC and sEMG signals to extract features relevant to gait analysis, enhancing the understanding of gait dynamics.
- Applied feature selection methods to recognize the important features that contribute to the accuracy of the gait parameter predictions.
- Employed ML techniques to predict temporal and spatial 17 gait parameters.
Following this introduction, the study organizes the paper into several key sections. The next section details the materials and methods employed in our study. The results and discussion section follows, providing an in-depth interpretation of the findings. The study concludes with a summary of these findings.
- The major contributions should be highlighted in enumerated form.
Response: Thank reviewer for your comments and suggestion. We add a paragraph to mention the contribution of this study. Line 98-103.
The main contribution of this study:
- Analyzed ACC and sEMG signals to extract features relevant to gait analysis, enhancing the understanding of gait dynamics.
- Applied feature selection methods to recognize the important features that contribute to the accuracy of the gait parameter predictions.
- Employed ML techniques to predict temporal and spatial 17 gait parameters.
- Add one paragraph about section wise organization of the paper at the end of introduction section.
Response: Thank reviewer for your comments and suggestion. We add a paragraph in section 1. Line 104-107.
Following this introduction, the study organizes the paper into several key sections. The next section details the materials and methods employed in our study. The results and discussion section follows, providing an in-depth interpretation of the findings. The study concludes with a summary of these findings.
- Literature survey part is missing. Add brief literature survey in the Introduction section.
Response: Thank reviewer for your comments and suggestion. We add the literature survey in the Introduction section our paper.
- Resolutions of Figure 1 and 2 need to be improved
Response: Thank reviewer for your comments and suggestion. We have addressed this comment and improved the Resolutions of Figure 1 and 2.
Figure 1. System architecture.
Figure 2. The block plot of the self-made multi-channel wireless sEMG and ACC measurement system including the slave and master boards. The slave board samples at 1000 Hz and the master board 500 Hz.
- Discuss gait parameters mentioned in section 2.3.1. in more detail
Response: Thank reviewer for your comments and suggestion. We add a table to mention 17 gait parameters in Table 1.
Table 1. Description of 17 temporospatial gait parameters.
|
Type |
Name |
Units |
Description |
|
Temporal Parameter
|
Heel-Strike (HS) |
Seconds |
The moment the heel makes contact with the ground. |
|
Gait Cycle Time (GCT) |
Steps/ minute |
Gait cycle duration is the time between heel strikes on the same foot. |
|
|
Double Leg Support (DS) |
% of cycle duration |
The bipedal stance period is when both feet are on the ground during the gait cycle. |
|
|
Cadence |
Steps/ minute |
The number of steps walked per minute. |
|
|
Stance |
% of cycle duration |
The foot hits the ground in the gait cycle's stance phase. |
|
|
Swing |
% of gait cycle |
The swing phase refers to the interval when the foot is not in contact with the ground during the gait cycle. |
|
|
Foot Flat Ratio (FFr) |
% of stance |
The phase of the stance in which the foot is completely in contact with the ground, with the sole entirely touching the surface. |
|
|
Push Ratio (Purify) |
% of stance |
The time between flat soles and lifted toes in stance. |
|
|
|
Load Ratio (LDr) |
% of stance |
The stance period from heel strike to full sole contact. |
|
Spatial Parameters |
Step Length |
meters |
The spatial distance between the feet when positioned on the ground. |
|
Stride Length |
meters |
Distance between heel strikes, equaling one gait cycle. |
|
|
Gait Speed |
meters/s |
Speed of forward walking. |
|
|
Peak Swing |
meters/s |
Maximum angular velocity from heel to toe during swing. |
|
|
Foot Pitch Angle at Heel Strike (HSP) |
degree |
The angle formed by the foot's contact with the ground upon impact. |
|
|
Foot Pitch Angle at Toe-Off (TOP) |
degree |
The angle of the toes relative to the ground just before lift-off at the end of the propulsion phase. |
|
|
Swing Width |
meters |
The largest sideways distance in the swing phase corresponds to the maximum lateral offset. |
|
|
3D Path Length |
% of stride length |
Depicts the scaled trajectory of the 3D gait cycle using stride length. |
- Why have authors used given signal parameters during feature extraction process? Specify reason
Response: Thank reviewer for your comments and suggestion. We use these parameters according to the previous studies with the sEMG signals [31,33,34,35]. Then, we add some sentences to mention these studies in section 2.3.2. Line 231-236
In our study, these parameters were carefully chosen for their relevance to signal characterization. MF and MDF are critical for understanding the spectral attributes and detecting muscle fatigue [33]. STD evaluates signal variability, offering insights into motor control [34][34]. SampEn measures data complexity, indicating physiological conditions’ predictability [31]. Lastly, RMS is utilized to assess the overall signal magnitude, which correlates with muscle activation levels [35].
- Elaborate feature selection procedure in more detail.
Response: Thank reviewer for your comments and suggestion. We add some sentences to mention the “feature selection” more clear in section 2.4. Line 238-245
2.4. Feature Selection
In this research, the XGBoost model was used to determine key features for predicting leg movements in gait parameters. XGBoost assesses feature importance by calculating scores to reflect each feature's utility in the model's decision trees. Features are evaluated based on frequency (how often they appear in trees), gain (impact on model accuracy), and coverage (number of data points a feature affects). These metrics help pinpoint the most influential features, aiding in model refinement and feature selection. Initially, the study extracted 84 features from the ACC and sEMG signals. Subsequently, a feature selection method was applied to identify the useful features for leg movement, resulting in the identification of 8 key features. Table 2 displays these eight important features used for estimating gait parameters.
- Why have authors preferred CatBoost, XGBoost and Decision tree classifiers only over other classifiers?
Response: Thank reviewer for your comments and suggestion. According the previous studies, these ML models have the higher results in the regression. We add some sentences to describe more clear in section 1. Line 87-93
This study focuses on examining the temporal and spatial parameters of walking patterns using sEMG and ACC data of lower limbs, analyzed through ML techniques such as Catboost, XGBoost and DT. This study selected CatBoost, XGBoost, and DT due to their strong performance in handling complex datasets, as similar studies [23][24][25]. CatBoost efficiently manages categorical data, while XGBoost is valued for its anti-overfitting capabilities. DT were chosen for their clear interpretability, which is essential in clinical applications. Together, these classifiers offer a balanced approach to our analysis.
- The accuracy in Table 4 should be mentioned in terms of percentage.
Response: Thank reviewer for your comments and suggestion. Table 4 has changed to Table 5. We have addressed this comment and changed the Table 5 accuracy in %.
Table 5. Models accuracy (R2 - Score) using ACC and sEMG features.
|
Models |
Accuracy (R2 - Score) |
|
Catboost |
91% |
|
XGBoost |
81% |
|
DT |
65% |
- Authors should use graphical representation of results during analysis of results.
Response: Thank reviewer for your comments and suggestion. We add graphical representations of all results in Figure 4. Line 358-372.
Table 4 displays a performance comparison of three different ML models: CatBoost, XGBoost, and DT, using various error metrics such as MSE, RMSE, MAE, and MAPE. CatBoost and XGBoost exhibit very similar performances, with CatBoost having a slight edge. For example, CatBoost shows an MSE of 7.65 and an RMSE of 1.49, indicating a lower average squared error, compared to XGBoost's MSE of 7.81 and RMSE of 1.53. Both models have a MAE of 1.00 and a MAPE of 0.03, suggesting strong predictive accuracy with low average and percentage errors. In contrast, the DT model underperforms, with higher error rates across all metrics: an MSE of 23.22, RMSE of 2.56, MAE of 1.49, and a MAPE of 0.04. These higher values indicate less accuracy in predicting outcomes compared to CatBoost and XGBoost. This study also calculates the accuracy (R2) shown in Table 5. Figure 4 shows the results of MSE (blue), RMSE (orange), MAE (light orange), MAPE (gray) and R2-Score (green) for the different models. The results show that CatBoost achieved a model accuracy (R2) of 91%, outperforming XGBoost, which has an 81% accuracy (R2). The DT model trails with 65% accuracy, underscoring its lower effectiveness in comparison to CatBoost and XGBoost.
Figure 4. The results of MSE (blue), RMSE (orange), MAE (light orange), MAPE (gray) and R2-Score (green) for CatBoost, XGboost and DT models.
- Add a table of existing works in the same domain and compare it with the proposed work in Results and Discussion section.
Response: Thank reviewer for your comments and suggestion. Due to the variations between previous studies and our own evaluation, we chose not to include a table. Instead, we extend the discussion section to include detailed comparisons. Line:418-432
This study also compares our findings with previous studies. Zhang et al. [22] used ML to estimate three gait parameters—stride length, velocity, and foot clearance—with MAEs of 2.37 ± 0.65, 3.10 ± 0.57, and 0.58 ± 0.28, respectively, for running. The MAE for 17 gait parameters in this study, predicted by CatBoost, is 1.00, which represents an improvement over the previous study. Moreover, Liu et al. only estimate 11 temporospatial gait parameters (six temporal parameters and five spatial parameters), PCC above 0.800, by sEMG signals [26]. In this study, the number of predicted gait parameters with PCC above 0.800 was fifteen when sEMG signals combining with ACC signals. In the another study, Wei et al. [27] utilized sEMG signals to recognize gait phases and predict lower limb kinematics. The research employed LDA, SVM, and LSTM as predictive models, collecting sEMG data from nine leg muscles to estimate four gait parameters: initial contact, foot flat, heel off, and toe off. The study found that LSTM achieved the highest performance with an accuracy of 93% and an RMSE of 1.86. In comparison, our study achieved a better RMSE of 1.46 and demonstrated superior PCC values. The results show that the IMU, ACC, could describe more detail leg-motion parameters when user walking or running.
- Highlight the limitations of the project work
Response: Thank reviewer for your comments and suggestion. We add the limitations of this study in section 5. Line 433-446
- Conclusions
This study marks a notable advancement in biomechanics, specifically in the analysis of leg movements, by integrating gait analysis with sEMG and ACC data. Utilizing advanced wireless technology, researchers gathered a dataset from 17 participants, enhancing our understanding of human locomotion. A significant aspect of this research is the use of a ML model to predict gait parameters from sEMG signals of thigh and calf muscles, along with ACC signals, demonstrating high accuracy. The CatBoost model's superiority over other models, such as XGBoost and Decision Tree, especially in processing complex high-frequency sEMG and ACC signals, is a key finding. This suggests potential for broader application in biomechanical data analysis. The study has limitations, including small datasets and challenges in improving the PCC results, as it could not accurately predict HS and Path Length. Future studies should consider enlarging the sample population and integrating neural network and deep learning approaches to achieve a more thorough and dependable analysis of gait.

Reviewer 2 Report
Comments and Suggestions for Authors
Please see the attached file

The quality of English can be overall slightly improved
Author Response
To Reviewer #2:
Thank the reviewer for his/her valuable comments that make better this manuscript. The texts in this revised manuscript have been corrected/ modified by yellow mark. It is our sincere hope that this revision will enhance readability and strengthen of the manuscript to satisfy the requirements of this prestigious journal.
Comments and Suggestions for Authors
Introduction
- Please explicit the gait parameters that you estimated in this study and add a brief but complete description of the state-of-the-art method to estimate them, focusing on the most widespread deterministic methods and ML methods. Which are the reasons to use a ML-based method with respect to the well-established methods for an accurate estimation of gait parameters?
Response: Thank reviewer for your comments and suggestion. We extend the Introduction section of our paper. Lines 27 -107
- Introduction
The gait, defined as the distinctive movement pattern of the lower extremities during ambulation, serves as a key indicator of the human body's locomotive attributes [1]. This biomechanical phenomenon is integral to the control mechanisms of prosthetic legs for lower limb amputees. Additionally, gait analysis plays a critical part in the realms of individual identification, fall risk assessment, and the diagnostic process for various disorders, including Parkinson's disease [2][3]. Traditional approaches to leg movement analysis, predominantly reliant on observational techniques and complex motion capture systems, provide substantial insights but are often constrained by practical, cost, and ecological validity factors.
Surface electromyography (sEMG) is instrumental in capturing the electrical activity of muscles during leg movement, providing invaluable data on muscle coordination and activation patterns [3]. In exploring the detailed interplay between neuromuscular system and physical movement, sEMGs play a crucial role [4]. They yield vital insights into the patterns of muscle activation, the robustness of muscular strength, and the dynamics of muscle exhaustion [5]. sEMG, an instrument essential for capturing the electrical fields emitted by muscle fibers upon contraction, utilizes motor unit action potentials (MUAPs) as its primary signal. These MUAPs, which represent the electrical patterns of muscle fibers, may increase in intensity or frequency based on the muscle's activity level. The sEMG methodology includes a system that amplifies and visually represents these electrical signals, thereby enabling researchers and clinicians to monitor muscle activation, timing, and coordination [6]. This capability is critically important for assessing muscle function, diagnosing neuromuscular disorders such as Myopathy, managing strength training, providing biomechanical insights, assisting in the design of orthotics and prosthetics, and advancing the development of ergonomic solutions [7]. sEMG in gait analysis proves beneficial for patients with neuromuscular conditions such as cerebral palsy, Parkinson's disease and muscle dystrophy [8]. Several researches have employed accelerometer (ACC) for the identification of gait parameters [9,10].
Recent advancements in gait analysis have focused on integrating inertial measurement unit (IMU) and sEMG data to enhance the estimation of gait parameters. These methods employ both types of sensors to capture comprehensive biomechanical movements, combining the kinematic data from IMUs with the kinetic data from sEMG sensors for a holistic view of gait dynamics. Notably, machine learning (ML) models namely, Random Forests (RF), Neural Networks (NNs), and Support Vector Machines (SVM) are increasingly utilized to analyze the rich, multidimensional data sets provided by these sensors. This approach not only improves the accuracy of gait parameter estimations but also facilitates real-time gait analysis, making it invaluable in clinical and sports settings for immediate feedback and continuous monitoring [11,12].
The use of ML methods facilitates adaptive learning from extensive datasets, which can enhance the accuracy and effectiveness of diagnostics and monitoring in clinical environments [13]. ML has demonstrated its efficiency in analyzing sEMG signals for various applications, including gesture classification [14], muscle fatigue detection [15][16], and identify sarcopenia [2]. In the previous research Kidziński et al. [17] employed a data-driven method to forecast the timings of foot-contact and foot-off events by analyzing marker time series and kinematic in children exhibiting both normal and abnormal walking patterns. Arunganesh et al. [18] applied a tree-based ML model to identify lower limb movements using sEMG measurements. In another study, Rastegari et al. [19] and Trabassi et al. [20] used ML models to identify Parkinson's disease through gait analysis based on ACC data. Howcroft et al. [21] utilized different ML techniques to estimate the risk of falls using ACC parameters. Zhang et al. [22] used Support Vector Regression (SVR) models to accurately estimate essential gait phases. Jani et al. [23] employed ML techniques such as CatBoost, Random Forest (RF), XGBoost, LightGBM, and Decision Tree (DT) for detecting gait abnormalities. The outcome of this study indicated that CatBoost achieved the highest accuracy compared to the other ML techniques. Wu et al. [24] used traditional ML such as XGBoost, RF, and Linear Discriminant Analysis (LDA) for the categorization of lower limb movements. Among these, XGBoost performed the best. Among these, XGBoost performed well. Armand et al. [25] used fuzzy decision trees to link clinical data with kinematic gait patterns of toe-walking. In the existing literature, only Liu et al. [26] have predicted 16 gait parameters, whereas other studies [27] [22] have typically predicted between 2 and 6 gait parameters.
This study focuses on examining the temporal and spatial parameters of walking patterns using sEMG and ACC data of lower limbs, analyzed through ML techniques such as CatBoost, XGBoost and DT. This study selected CatBoost, XGBoost, and DT due to their strong performance in handling complex datasets, as similar studies [23][24][25]. CatBoost efficiently manages categorical data, while XGBoost is valued for its anti-overfitting capabilities. DT were chosen for their clear interpretability, which is essential in clinical applications. Together, these classifiers offer a balanced approach to our analysis. The sEMG data were collected from two primary muscles in each foot, the vastus lateralis and gastrocnemius muscles. ACC were placed at the thighs. For benchmarking, the study utilized the GaitUp Physilog®5 shoe-worn inertial sensors, a professional gait analysis tool [28]. A wireless self-made multi-channel measurement system was employed to record muscle activity and thigh motion during treadmill running. The main contribution of this study:
- Analyzed ACC and sEMG signals to extract features relevant to gait analysis, enhancing the understanding of gait dynamics.
- Applied feature selection methods to recognize the important features that contribute to the accuracy of the gait parameter predictions.
- Employed ML techniques to predict temporal and spatial 17 gait parameters.
Following this introduction, the study organizes the paper into several key sections. The next section details the materials and methods employed in our study. The results and discussion section follows, providing an in-depth interpretation of the findings. The study concludes with a summary of these findings.
- Add a state-of-the-art description of the methods that have been proposed to estimate gait parameters exploiting both inertial and emg data.
Response: Thank reviewer for your comments and suggestion. We add literature in the Introduction section of our paper. Lines 76 -86.
- Which is the lack in the literature that this study can fill?
Response: Thank reviewer for your comments and suggestion. We add literature in the Introduction section of our paper. Lines 76 -86.
The use of ML methods facilitates adaptive learning from extensive datasets, which can enhance the accuracy and effectiveness of diagnostics and monitoring in clinical environments [13]. ML has demonstrated its efficiency in analyzing sEMG signals for various applications, including gesture classification [14], muscle fatigue detection [15][16], and identify sarcopenia [2]. In the previous research Kidziński et al. [17] employed a data-driven method to forecast the timings of foot-contact and foot-off events by analyzing marker time series and kinematic in children exhibiting both normal and abnormal walking patterns. Arunganesh et al. [18] applied a tree-based ML model to identify lower limb movements using sEMG measurements. In another study, Rastegari et al. [19] and Trabassi et al. [20] used ML models to identify Parkinson's disease through gait analysis based on ACC data. Howcroft et al. [21] utilized different ML techniques to estimate the risk of falls using ACC parameters. Zhang et al. [22] used Support Vector Regression (SVR) models to accurately estimate essential gait phases. Jani et al. [23] employed ML techniques such as CatBoost, Random Forest (RF), XGBoost, LightGBM, and Decision Tree (DT) for detecting gait abnormalities. The outcome of this study indicated that CatBoost achieved the highest accuracy compared to the other ML techniques. Wu et al. [24] used traditional ML such as XGBoost, RF, and Linear Discriminant Analysis (LDA) for the categorization of lower limb movements. Among these, XGBoost performed the best. Among these, XGBoost performed well. Armand et al. [25] used fuzzy decision trees to link clinical data with kinematic gait patterns of toe-walking. In the existing literature, only Liu et al. [26] have predicted 11 temporospatial gait parameters, whereas other studies [27] [22] have typically predicted between 2 and 6 gait parameters.
Materials and methods
- Explicit also in this section the activities of which muscles are recorded.
Response: Thank reviewer for your comments and suggestion. We add sentences to mention the experiment. Line 152-165.
2.2. Experiment Protocol
The study involved voluntary participants who were adults diagnosed with sarcopenia, had healthy limbs, and maintained normal standing positions. The group included 17 individuals aged 19 to 23, with an average age of 20 ± 1 years. The average height of the individuals was 156 ± 4.6 cm, with a range of 149 to 164 cm. Similarly, the average weight was 45.9 ± 5.7 kg, with a range of 31 to 56 kg. The average shoe size among participants was 23.9±0.6, varying from 23 to 25 cm. The four electrodes were placed on the gastrocnemius and vastus lateralis muscles of left and right legs for sEMG measurements, the ACC was placed on the outer thigh for recoding the thigh motion, and the GaitUp Physilog® [28–30] shoe-worn inertial sensors were positioned on the tongues of shoes for gait status measurement. Participants evaluated their own physical health before participating in the experiment. The Chung Shan University Hospital's Research Ethics Committee in Taichung city, Taiwan, approved this experiment with the reference number CS2-22210.
- 3: add units in the y-axis.
Response: Thank reviewer for your comments and suggestion. We add units in the y-axis in Figure 3.
Figure 3. sEMGs and ACC signals under walking.
- Please rephrase the sentence at lines 114-115: how was performed the data segmentation?
Response: Thank reviewer for your comments and suggestion. We add some sentences to mention the data process more clear. Line 166-178.
2.3. Data Processing
The study recorded data for 6 minutes, totaling 180,000 data points for both sEMGs and ACCs. Data segmentation started with intentional muscle contractions in sEMG recordings and activity onset in the GaitUp Lab, marking the beginning of data slicing. This step is critical as it marks the point where significant gait events begin to be recorded, ensuring that the analysis focuses on relevant data. To ensure the integrity of our analysis and mitigate the impact of potential outliers or anomalies at the start and end of each gait cycle, we strategically omitted 7.5 seconds from both the start and the end of the signal. The data, including two-channel sEMGs, x- and y-axis signals of ACC and GaitUp Physilog® 5 sensor readings, was divided into 30-second batches with a 15-second window shift, resulting in 22 sample sets per experiment. Gait parameters were derived by segmenting and processing the data captured by GaitUp Physilog® 5 using the GaitUp Lab software, which were the target outputs.
- Please add the used definitions of swing width and path length, since they are not unique.
Response: Thank reviewer for your comments and suggestion. We add a table in section 2.3.1 to describe the 17 temporospatial gait parameters.
Table 1. Description of 17 temporospatial gait parameters.
|
Type |
Name |
Units |
Description |
|
Temporal Parameter
|
Heel-Strike (HS) |
Seconds |
The moment the heel makes contact with the ground. |
|
Gait Cycle Time (GCT) |
Steps/ minute |
Gait cycle duration is the time between heel strikes on the same foot. |
|
|
Double Leg Support (DS) |
% of cycle duration |
The bipedal stance period is when both feet are on the ground during the gait cycle. |
|
|
Cadence |
Steps/ minute |
The number of steps walked per minute. |
|
|
Stance |
% of cycle duration |
The foot hits the ground in the gait cycle's stance phase. |
|
|
Swing |
% of gait cycle |
The swing phase refers to the interval when the foot is not in contact with the ground during the gait cycle. |
|
|
Foot Flat Ratio (FFr) |
% of stance |
The phase of the stance in which the foot is completely in contact with the ground, with the sole entirely touching the surface. |
|
|
Push Ratio (Purify) |
% of stance |
The time between flat soles and lifted toes in stance. |
|
|
|
Load Ratio (LDr) |
% of stance |
The stance period from heel strike to full sole contact. |
|
Spatial Parameters |
Step Length |
meters |
The spatial distance between the feet when positioned on the ground. |
|
Stride Length |
meters |
Distance between heel strikes, equaling one gait cycle. |
|
|
Gait Speed |
meters/s |
Speed of forward walking. |
|
|
Peak Swing |
meters/s |
Maximum angular velocity from heel to toe during swing. |
|
|
Foot Pitch Angle at Heel Strike (HSP) |
degree |
The angle formed by the foot's contact with the ground upon impact. |
|
|
Foot Pitch Angle at Toe-Off (TOP) |
degree |
The angle of the toes relative to the ground just before lift-off at the end of the propulsion phase. |
|
|
Swing Width |
meters |
The largest sideways distance in the swing phase corresponds to the maximum lateral offset. |
|
|
3D Path Length |
% of stride length |
Depicts the scaled trajectory of the 3D gait cycle using stride length. |
- Add a reference of a study assessing the methods used by GaitUp Lab and its accuracy.
Response: Thank reviewer for your comments and suggestion. We add a reference [28] to show the validation of Shoe-Worn Gait Up Physilog®5 Wearable Inertial Sensors.
[28] Carroll, K.; Kennedy, R.A.; Koutoulas, V.; Bui, M.; Kraan, C.M. Validation of Shoe-Worn Gait Up Physilog®5 Wearable Inertial Sensors in Adolescents. Gait Posture 2022, 91, 19–25, doi:10.1016/j.gaitpost.2021.09.203.”
- Statistical analysis: is the GaitUp Lab can be considered a gold stardand? If not, I suggest to not define ‘errors’ the differences between the predicted and actual values, but indeed only differences.
Response: Thank reviewer for your comments and suggestion. In this study, we used the GaitUp Lab as the standard because the previous studies [28,29,30] have validated the performance of GaitUp Lab.
- Carroll, K.; Kennedy, R.A.; Koutoulas, V.; Bui, M.; Kraan, C.M. Validation of Shoe-Worn Gait Up Physilog®5 Wearable Inertial Sensors in Adolescents. Gait Posture 2022, 91, 19–25, doi:10.1016/j.gaitpost.2021.09.203.
- Dadashi, F.; Mariani, B.; Rochat, S.; Büla, C. J.; Santos-Eggimann, B.; Aminian, K. Gait and Foot Clearance Parameters Obtained Using Shoe-Worn Inertial Sensors in a Large-Population Sample of Older Adults. Sensors 2014, 14, 443-457, doi:10.3390/s140100443
- Homan, K.; Yamamoto, K.; Kadoya, K.; Ishida, N.; Iwasaki, N. Comprehensive Validation of a Wearable Foot Sensor System for Estimating Spatiotemporal Gait Parameters by Simultaneous Three-Dimensional Optical Motion Analysis. BMC Sports Sci. Med. Rehabil. 2022, 14, 71, doi:10.1186/s13102-022-00461-x.
- Statistical analysis: please also mention and explicit the computation of the models accuracy.
Response: Thank reviewer for your comments and suggestion. We add some sentences to mention the accuracy of model in section 2.6. Line 302-316.
The R² score, also known as the coefficient of determination, quantifies how much of the variance in the dependent variable that is explained by the independent variables in a regression model. It is a crucial indicator of model fit, with a score of 1 signifying perfect prediction accuracy.
(13)
where:
- SSR (Sum of Squares of Residuals) calculates the sum of the squared differences between the observed values and the values predicted by the model. It is mathematically represented as:
, where are the actual values and denotes the prediction values.
- TSS (Total Sum of Squares) represents the overall variance within the dataset, determined by summing the squared deviations of each observed value from the dataset's mean:
, where represents the observed value mean.
Results and discussion
- Table 2: you meant stride length instead of strike length? Please correct the parameters names.
Response: Thank reviewer for your comments and suggestion. We correct ‘strike length’ to ‘stride length’
- Table 3: which are the units of the performance metrics? Please add the errors you obtained for each gait parameter with its proper unit.
Response: Thank reviewer for your comments and suggestion. Response: We add a table to describe the 17 temporospatial gait parameters and their units in section 2.3.1. Table 3 is changed to Table 4. In Table 4, the performances metrics shows the total error of 17 temporospatial parameters. Thus, we could not give these errors with the units.
- Please add a proper Discussion section instead of merging results and discussion together. You should investigate more thoroughly in this section: 1) a comparison among the implemented methods and the explicit suggestion of the best of them; 2) a comment on which results you expected from subject with sarcopenia based on the relevant previous literature; 3) a complete comparison between your results and the results from the previous works (both deterministic and ML-based); 4) limitations of the study.
Response: Thank reviewer for your comments and suggestion. We modify the texts of Discussion, and add the limitations of this study in Conclusions.
Line 386-430
- Discussion
The gait analysis is a finite element analysis, which is a key component of modern therapeutic and rehabilitative programs for observing patients’ walking. Pheasant et al. [44] presented “Human Walking” whom were the first to study gait kinematics using video camera techniques for human walking. This method captures the kinematic dynamics in the spatial and temporal domains when subjects walking [45]. The sEMG signals of subjects are also measuring to explore the muscles’ conditions. The limitation of the video camera technique is that subjects will be measured at a motion laboratory. Now, the IMUs, like as accelerometer and gyroscope, are used to measure the temporal-spatial parameters of gaits [28]. The advantage of the IMUs is that subjects can naturally walk on the ground. However, this method also needs to combine the sEMG measurement to understand the muscle activities. Muro-de-la-Herran et al. [46] reviewed publications from 2012 and 2013 related to gait analysis techniques and their clinical applications. Out of the thirty-two articles they found, 40% discussed video camera systems, 37.5% dealt with systems based on IMUs, and 22.5% examined various other wearable sensor systems. This study used sEMG and ACC to predict the 17 temporal-spatial parameters of gaits, and approach to 0.91 of accuracy (R2). Our proposed method uses sEMG and IMUs to predict gait parameters. Out of 17 parameters, 15 were accurately predicted by the CatBoost model, which PCCs of left and right feet were larger than 0.800. At the same time, the sEMG signals provide valuable information about muscle activities. In this study, two gait parameters—3D Path Length and HS—showed lower PCC values compared to others. The PCCs of Path Length for the left and right feet have the very large difference (0.24 vs 0.83). But, in the previous study [26], PCCs of Path Length were 0.90 and 0.90 for left and right feet with XGBoost. This result showed that the features of ACC could not be used to predict the Path Length. According to Cruz-Jentoft et al. [47], sarcopenia typically leads to reductions in muscle mass and strength, which are correlated with slower walking speeds, shorter stride lengths, and increased gait variability. These outcomes reflect the muscle weakening associated with the condition and its impact on the functionality and stability of the lower limbs during movement. Therefore, this method provides more comprehensive insights into gait analysis. These gait parameters are crucial for diagnosing gait disorders and designing assistive devices as well as the customization of prosthetics and orthotics.
This study also compares our findings with previous studies. Zhang et al. [22] used ML to estimate three gait parameters—stride length, velocity, and foot clearance—with MAEs of 2.37 ± 0.65, 3.10 ± 0.57, and 0.58 ± 0.28, respectively, for running. The MAE for 17 gait parameters in this study, predicted by CatBoost, is 1.00, which represents an improvement over the previous study. Moreover, Liu et al. only estimate 11 temporospatial gait parameters (six temporal parameters and five spatial parameters), PCC above 0.800, by sEMG signals [26]. In this study, the number of predicted gait parameters with PCC above 0.800 was fifteen when sEMG signals combining with ACC signals. In the another study, Wei et al. [27] utilized sEMG signals to recognize gait phases and predict lower limb kinematics. The research employed LDA, SVM, and LSTM as predictive models, collecting sEMG data from nine leg muscles to estimate four gait parameters: initial contact, foot flat, heel off, and toe off. The study found that LSTM achieved the highest performance with an accuracy of 93% and an RMSE of 1.86. In comparison, our study achieved a better RMSE of 1.46 and demonstrated superior PCC values. The results show that the IMU, ACC, could describe more detail leg-motion parameters when user walking or running.
Line 431-444
- Conclusions
This study marks a notable advancement in biomechanics, specifically in the analysis of leg movements, by integrating gait analysis with sEMG and ACC data. Utilizing advanced wireless technology, researchers gathered a dataset from 17 participants, enhancing our understanding of human locomotion. A significant aspect of this research is the use of a ML model to predict gait parameters from sEMG signals of thigh and calf muscles, along with ACC signals, demonstrating high accuracy. The CatBoost model's superiority over other models, such as XGBoost and Decision Tree, especially in processing complex high-frequency sEMG and ACC signals, is a key finding. This suggests potential for broader application in biomechanical data analysis. The study has limitations, including small datasets and challenges in improving the PCC results, as it could not accurately predict HS and Path Length. Future studies should consider enlarging the sample population and integrating neural network and deep learning approaches to achieve a more thorough and dependable analysis of gait.
- Since the most relevant feature are the ones coming from the accelerations, is it necessary to use the emg? Please add the results of the methods only considering the acceleration based features.
Response: Thank reviewer for your comments and suggestion. We add Table 6 which shows results only considering the acceleration based features. PCC for 17 temporospatial gait parameters of left and right feet using CatBoost are 0.846 ± 0.174 (mean ± STD) for the left and 0.886 ± 0.40 for the right feet. Comparing these results with those obtained using a combination of ACC and sEMG features, it was evident that the combined features yielded better outcomes. Line 373-377
This study also focused on predicting gait parameters using only ACC features, as a shown in Table 6. For this purpose, the CatBoost model was employed. The resulting mean ± STD values were 0.846 ± 0.174 for the left and 0.886 ± 0.40 for the right feet. Comparing these results with those obtained using a combination of ACC and sEMG features, it was evident that the combined features yielded better outcomes.
Table 6. PCC for 17 temporospatial gait parameters of left and right feet using CatBoost with ACC features.
|
Name |
L |
R |
|
HS |
0.69 |
0.81 |
|
GCT |
0.90 |
0.92 |
|
Cadence |
0.88 |
0.90 |
|
Stance |
0.94 |
0.90 |
|
Swing |
0.94 |
0.90 |
|
LDr |
0.88 |
0.86 |
|
FFr |
0.89 |
0.89 |
|
PUr |
0.89 |
0.90 |
|
DS |
0.93 |
0.93 |
|
Stride Length |
0.89 |
0.90 |
|
Gait Speed |
0.78 |
0.82 |
|
Peak Swing |
0.93 |
0.93 |
|
HSP |
0.92 |
0.89 |
|
TOP |
0.94 |
0.94 |
|
Swing Width |
0.90 |
0.92 |
|
3D Path Length |
0.22 |
0.83 |
|
Step Length |
0.84 |
0.84 |
|
Mean±STD |
0.846 ± 0.174 |
0.886 ± 0.40 |
- How can you explain a Pearson correlation higher than 0.9 in the stride length and lower than 0.3 in path length in left side? Why is there always a difference between left and right side results in terms of path length?
Response: Thank reviewer for your comments and suggestion. The PCCs of Path Length for the left and right feet have the very large difference (0.24 vs 0.83). But, in the previous study [26], PCCs of Path Length were 0.90 and 0.90 for left and right feet with XGBoost. This result showed that the features of ACC could not be used to predict the Path Length. We add these mentions in texts of Discussion. Line 385-416.
- Discussion
The gait analysis is a finite element analysis, which is a key component of modern therapeutic and rehabilitative programs for observing patients’ walking. Pheasant et al. [44] presented “Human Walking” whom were the first to study gait kinematics using video camera techniques for human walking. This method captures the kinematic dynamics in the spatial and temporal domains when subjects walking [45]. The sEMG signals of subjects are also measuring to explore the muscles’ conditions. The limitation of the video camera technique is that subjects will be measured at a motion laboratory. Now, the IMUs, like as accelerometer and gyroscope, are used to measure the temporal-spatial parameters of gaits [28]. The advantage of the IMUs is that subjects can naturally walk on the ground. However, this method also needs to combine the sEMG measurement to understand the muscle activities. Muro-de-la-Herran et al. [46] reviewed publications from 2012 and 2013 related to gait analysis techniques and their clinical applications. Out of the thirty-two articles they found, 40% discussed video camera systems, 37.5% dealt with systems based on IMUs, and 22.5% examined various other wearable sensor systems. This study used sEMG and ACC to predict the 17 temporal-spatial parameters of gaits, and approach to 0.91 of accuracy (R2). Our proposed method uses sEMG and IMUs to predict gait parameters. Out of 17 parameters, 15 were accurately predicted by the CatBoost model, which PCCs of left and right feet were larger than 0.800. At the same time, the sEMG signals provide valuable information about muscle activities. In this study, three gait parameters—3D Path Length and HS—showed lower PCC values compared to others. The PCCs of Path Length for the left and right feet have the very large difference (0.24 vs 0.83). But, in the previous study [26], PCCs of Path Length were 0.90 and 0.90 for left and right feet with XGBoost. This result showed that the features of ACC could not be used to predict the Path Length. According to Cruz-Jentoft et al. [47], sarcopenia typically leads to reductions in muscle mass and strength, which are correlated with slower walking speeds, shorter stride lengths, and increased gait variability. These outcomes reflect the muscle weakening associated with the condition and its impact on the functionality and stability of the lower limbs during movement. Therefore, this method provides more comprehensive insights into gait analysis. These gait parameters are crucial for diagnosing gait disorders and designing assistive devices as well as the customization of prosthetics and orthotics.

Reviewer 3 Report
Comments and Suggestions for Authors
The authors present a study about predicting the gait parameters using sEMG and accelerometer sensors using a machine learning algorithm. The system consists of a hardware part with multiple measurement points and a software processing part.
The manuscript describes the hardware architecture quite short, the parameters to be calculated, the statistical parameters used to evaluate the results and some experimental results and discussions. The conclusions are also quite superficial.
Please find below some remarks about things to be improved in the present manuscript
-
As I understand the data is acquired with 1000 Hz from the slave node and it reaches the PC through the master node which has a lower communication speed, doubled the its own data. How is done the transmission without loosing data?
-
In fig 3. - how are the X and Y axes of the ACC oriented with respect to the leg?
-
I suggest to detail a bit these parameters (Heel-Strike (HS), Gait Cycle Time (gct), Double Leg 125 Support (DS), Cadence, Stance and Swing Ratios, Load Ratio (LDr), Foot Flat Ratio (FFr), 126 and Push Ratio (Purify)) in term of math and signification.
-
“D signifies the quantity of discrete Fourier transform” - what means “the quantity”?
-
Eq 3 - xi - are the samples in the time domain?
-
In the case of Entropy: the variables presented need to be more detailed. How are chosen the values for m and v . Who is Z’ in eq. 7? In eq 8 and 9 which are the limits for i and j? What does it mean d[Xm(i), Xm(j)]?
-
In eq 14 - The authors said that a and b are vectors. What means a-ma and b-mb? What means SUM(a-ma)(b-mb)
-
“which has been studies for the clinical” - studied instead of studied
-
“was the first studied the gait kinematics by the video camera technique” - the first which studied…
-
“the research gathered an extensive dataset from 17 participants” - data from these 17 participants are not presented
-
“In Table 2, the models provided estimates for 17 gait parameters for both the left and right feet.” - what are the values fed into the algorithms and what does mean "the estimate" ?
-
How is computed the accuracy in table 4?
-
Some parameters are computed, but what they reveal about the body is unclear.
-
“development of a ML model to predict gait parameters” - which is the developed algorithm? As I understand you use an algorithm already available. Please detail what means “the development”!
-
The introduction must be improved. You have to offer more details about sEMG working principle, so the reader can understand how the signals are connected to the muscle activities.
-
Give more details about the used algorithms Catboost, XGBoost, and DT.
- Improve the Conclusions. May be a discussion in respect with other methods presented in the literature would help.
- Detail the utility of these parameters you estimate.
The authors must give a bit more attention to the text.
Author Response
To Reviewer #3:
Thank the reviewer for his/her valuable comments that make better this manuscript. The texts in this revised manuscript have been corrected/ modified by yellow mark. It is our sincere hope that this revision will enhance readability and strengthen of the manuscript to satisfy the requirements of this prestigious journal.
Please find below some remarks about things to be improved in the present manuscript
- As I understand the data is acquired with 1000 Hz from the slave node and it reaches the PC through the master node which has a lower communication speed, doubled the its own data. How is done the transmission without loosing data?
Response: Thank reviewer for your comments and suggestion. We used two different sampling rates to avoid the transmission without loosing data. We add some sentences to mention more clear in section 2.1. Line 122-138.
2.1. sEMG and Accelerometer Measuring Device
This study developed a self-made multi-channel wireless sEMG and ACC measurement system, as illustrated in Figure 2. The system comprises two boards, namely a slave and a master, that utilize XBEE S2C modules for the purpose of data transfer between two boards. They are placed at the left and right legs to measure the sEMGs of the thigh and calf muscles and mention the thigh motion by the x and y axes of the ACC. The sampling rate on the slave board is 1000 Hz, and the transfer rate is also 1000 Hz. The master board is the 500 Hz of sampling, which includes an HC-05 Bluetooth module to send sEMG and ACC data from slave and master boards to the PC at the frequency of 500 Hz. Dual-channel sEMG and dual-channel ACC signals equip one board. The sEMG circuitry is designed following Liu et al.'s study [16]. The system operates on a Texas Instruments (TI) MSP430F5438A microcontroller with a 12-bit analog to digital (ADC) resolution. Data packets to the PC have 2 header bytes and 20 data bytes. In order to prevent data loss from the slave board to the PC terminal, we employed a down-sampling method. The slave board transmitted data to the master board at 1000 Hz, and the master board then transmitted data to the PC terminal at 500 Hz. We found that this method effectively prevented any data loss from the slave board.
- In fig 3. - how are the X and Y axes of the ACC oriented with respect to the leg?
Response: Thank reviewer for your comments and suggestion. In Fig. 3, we can find that x-axis signal of ACC is significantly oriented toward the calf muscle active. We add one sentence to describe it in section 2.1. Line 122-138.
Figure 3 displays the temporal progression of the four sEMG signals, and x and y axes of two ACCs recorded using the designed device during walking, covering a time span of 6 seconds. The activities of thigh and calf muscles of left and right legs show the time sequences. the x and y axes of the accelerometers (ACCs) on the left and right legs exhibit a similar phenomenon, where the x-axis signal is significantly oriented toward the calf muscle active. Thus, we hypothesize that the gait status could be described by these signals.
- I suggest to detail a bit these parameters (Heel-Strike (HS), Gait Cycle Time (gct), Double Leg 125 Support (DS), Cadence, Stance and Swing Ratios, Load Ratio (LDr), Foot Flat Ratio (FFr), 126 and Push Ratio (Purify)) in term of math and signification.
Response: Thank reviewer for your comments and suggestion. We add Table 1 for description of 17 temporospatial gait parameters. Line 180-192.
2.3.1. Gait Parameters
The GaitUp Lab gait analysis system assessed 17 left and right foot gait parameters, including 9 temporal and 8 spatial parameters. Temporal parameters include Heel-Strike (HS), Gait Cycle Time (gct), Double Leg Support (DS), Cadence, Stance and Swing Ratios, Foot Flat Ratio (FFr), Push Ratio (Purify) and, Load Ratio (LDr). These parameters measure aspects such as the heel's contact with the ground, the duration and phases of the gait cycle, and weight distribution.
Spatial parameters include Stride Length, Step Length, Gait Speed, Peak Swing (maximum leg angular velocity during the swing phase), Foot Pitch Angles at Heel Strike (HSP) and Toe-Off (TOP), Swing Width, and 3D Path Length. These focus on the distance, speed, and angles involved in walking movements. This categorization offers a detailed view of the dynamics of human gait. The gait parameters described in Table 1.
Table 1. Description of 17 temporospatial gait parameters.
|
Type |
Name |
Units |
Description |
|
Temporal Parameter
|
Heel-Strike (HS) |
Seconds |
The moment the heel makes contact with the ground. |
|
Gait Cycle Time (GCT) |
Steps/ minute |
Gait cycle duration is the time between heel strikes on the same foot. |
|
|
Double Leg Support (DS) |
% of cycle duration |
The bipedal stance period is when both feet are on the ground during the gait cycle. |
|
|
Cadence |
Steps/ minute |
The number of steps walked per minute. |
|
|
Stance |
% of cycle duration |
The foot hits the ground in the gait cycle's stance phase. |
|
|
Swing |
% of gait cycle |
The swing phase refers to the interval when the foot is not in contact with the ground during the gait cycle. |
|
|
Foot Flat Ratio (FFr) |
% of stance |
The phase of the stance in which the foot is completely in contact with the ground, with the sole entirely touching the surface. |
|
|
Push Ratio (Purify) |
% of stance |
The time between flat soles and lifted toes in stance. |
|
|
|
Load Ratio (LDr) |
% of stance |
The stance period from heel strike to full sole contact. |
|
Spatial Parameters |
Step Length |
meters |
The spatial distance between the feet when positioned on the ground. |
|
Stride Length |
meters |
Distance between heel strikes, equaling one gait cycle. |
|
|
Gait Speed |
meters/s |
Speed of forward walking. |
|
|
Peak Swing |
meters/s |
Maximum angular velocity from heel to toe during swing. |
|
|
Foot Pitch Angle at Heel Strike (HSP) |
degree |
The angle formed by the foot's contact with the ground upon impact. |
|
|
Foot Pitch Angle at Toe-Off (TOP) |
degree |
The angle of the toes relative to the ground just before lift-off at the end of the propulsion phase. |
|
|
Swing Width |
meters |
The largest sideways distance in the swing phase corresponds to the maximum lateral offset. |
|
|
3D Path Length |
% of stride length |
Depicts the scaled trajectory of the 3D gait cycle using stride length. |
- “D signifies the quantity of discrete Fourier transform” - what means “the quantity”?
Response: Thank reviewer for your comments and suggestion. D represents the number of discrete Fourier transform. We modify this description. Line:200
- Eq 3 - xi - are the samples in the time domain?
Response: Thank reviewer for your comments and suggestion. Yes, in Eq 3, xi is the sample data in the time domain. N is the size of window. We modify the description for Eq. 3. Line 205-207.
, (3)
where xi represents the sampled data, represents mean of x, and N denotes the size of window.
- In the case of Entropy: the variables presented need to be more detailed. How are chosen the values for m and v . Who is Z’ in eq. 7? In eq 8 and 9 which are the limits for i and j? What does it mean d[Xm(i), Xm(j)]?
Response: Thank reviewer for your comments and suggestion. We modify the description of Sample Entropy. Line 210-227
Sample Entropy (SampEn) is used to calculate the intricacy and regularity of a signal's temporal sequence. An increased entropy value indicates time series complexity. Calculating sample entropy requires defining dimension m and selecting a suitable v value. After determining m, segment signal labeled Xm. The method is outlined as stated below,
. (5)
Equation (6) is the formula for SampEn.
, (6)
where,
(7)
. (8)
The function measures the distance between vectors and in the reconstructed phase space, where each vector represents a subsequence of length starting at points i and j respectively. The limits for i and j in Equations (7) and (8) are intended to span the entire dimension of the time series, ensuring that all possible pairs of subsequences of length m and m+1 are considered in the computation. Typically, i and j range from 1 to S-m or S-m+1. This research sets m=2 and v = STD × 0.2. The variables m and v in our SampEn calculations are chosen based on the raw data and the specific criteria of our analysis, following the methodology of previous studies [31,32].
- In eq 14 - The authors said that a and b are vectors. What means a-ma and b-mb? What means SUM(a-ma)(b-mb)
Response: Thank reviewer for your comments and suggestion. We modify the description of Eq. (14). Line 320-329.
We present the numerical data as a mean (M) ± standard deviation (STD). To elucidate the connection between the target and predicted values observed in our test dataset, we have employed the Pearson correlation coefficient (PCC). The mathematical formulation of this coefficient is explicitly detailed in Equation (14), wherein ma and mb corresponds to the mean values of the dataset a and b, respectively. Thus, a−ma​ is the deviation of a single value ‘a’ from its mean, ma​. Likewise, b−mb​ is the deviation of a single value ‘b’ from its mean, mb​. In the PCC equation represents the sum of the products of the deviations of each pair of corresponding values from their respective means. This statistical approach enables a rigorous evaluation of the linear correlation between the two sets of variables under investigation.
. (14)
- “which has been studies for the clinical” - studied instead of studied.
Response: Thank reviewer for your comments and suggestion. We modify this paragraph to describe more clear. Line 385-415.
The gait analysis is a finite element analysis, which is a key component of modern therapeutic and rehabilitative programs for observing patients’ walking. Pheasant et al. [44] presented “Human Walking” whom were the first to study gait kinematics using video camera techniques for human walking. This method captures the kinematic dynamics in the spatial and temporal domains when subjects walking [45]. The sEMG signals of subjects are also measuring to explore the muscles’ conditions. The limitation of the video camera technique is that subjects will be measured at a motion laboratory. Now, the IMUs, like as accelerometer and gyroscope, are used to measure the temporal-spatial parameters of gaits [28]. The advantage of the IMUs is that subjects can naturally walk on the ground. However, this method also needs to combine the sEMG measurement to understand the muscle activities. Muro-de-la-Herran et al. [46] reviewed publications from 2012 and 2013 related to gait analysis techniques and their clinical applications. Out of the thirty-two articles they found, 40% discussed video camera systems, 37.5% dealt with systems based on IMUs, and 22.5% examined various other wearable sensor systems. This study used sEMG and ACC to predict the 17 temporal-spatial parameters of gaits, and approach to 0.91 of accuracy (R2). Our proposed method uses sEMG and IMUs to predict gait parameters. Out of 17 parameters, 15 were accurately predicted by the CatBoost model, which PCCs of left and right feet were larger than 0.800. At the same time, the sEMG signals provide valuable information about muscle activities. In this study, three gait parameters—3D Path Length and HS—showed lower PCC values compared to others. The PCCs of Path Length for the left and right feet have the very large difference (0.24 vs 0.83). But, in the previous study [26], PCCs of Path Length were 0.90 and 0.90 for left and right feet with XGBoost. This result showed that the features of ACC could not be used to predict the Path Length. According to Cruz-Jentoft et al. [47], sarcopenia typically leads to reductions in muscle mass and strength, which are correlated with slower walking speeds, shorter stride lengths, and increased gait variability. These outcomes reflect the muscle weakening associated with the condition and its impact on the functionality and stability of the lower limbs during movement. Therefore, this method provides more comprehensive insights into gait analysis. These gait parameters are crucial for diagnosing gait disorders and designing assistive devices as well as the customization of prosthetics and orthotics.
- “was the first studied the gait kinematics by the video camera technique” - the first which studied.
Response: Thank reviewer for your comments and suggestion. We modify this sentence. “ Pheasant et al. [44] presented “Human Walking” whom were the first to study gait kinematics using video camera techniques for human walking.”
- “the research gathered an extensive dataset from 17 participants” - data from these 17 participants are not presented.
Response: Thank reviewer for your comments and suggestion. We add some sentences to describe the information of subjects in section 2.2. Line 152-165.
2.2. Experiment Protocol
The study involved voluntary participants who were adults diagnosed with sarcopenia, had healthy limbs, and maintained normal standing positions. The group included 17 individuals aged 19 to 23, with an average age of 20 ± 1 years. The average height of the individuals was 156 ± 4.6 cm, with a range of 149 to 164 cm. Similarly, the average weight was 45.9 ± 5.7 kg, with a range of 31 to 56 kg. The average shoe size among participants was 23.9±0.6, varying from 23 to 25 cm. The four electrodes were placed on the gastrocnemius and vastus lateralis muscles of left and right legs for sEMG measurements, the ACC was placed on the outer thigh for recoding the thigh motion, and the GaitUp Physilog® [28–30] shoe-worn inertial sensors were positioned on the tongues of shoes for gait status measurement. Participants evaluated their own physical health before participating in the experiment. The Chung Shan University Hospital's Research Ethics Committee in Taichung city, Taiwan, approved this experiment with the reference number CS2-22210.
- “In Table 2, the models provided estimates for 17 gait parameters for both the left and right feet.” - what are the values fed into the algorithms and what does mean "the estimate"?
Response: Thank reviewer for your comments and suggestion. Table 2 is changed to Table 3. We modify the descriptions. Line 331-338.
- Results
The study employed CatBoost, XGBoost, and DT algorithms to predict gait parameters. Features extracted from the parameters of signals served as input variables, with the GaitUp value used as the target variable. The dataset was separated into training (80%) and testing (20%) subsets, with the testing dataset used to evaluate the trained models. Table 3 shows the PCCs for predicting 17 left and right foot gait parameters. Where CatBoost and XGBoost predict 15 gait parameters correctly with a PCC value above 0.80 for both feet.
- How is computed the accuracy in table 4?
Response: Thank reviewer for your comments and suggestion. To compute the accuracy of regression model, we have used R2. We add some sentences to mention Eq. (13) in section 2.6. Line: 302-316
The R² score, also known as the coefficient of determination, quantifies how much of the variance in the dependent variable that is explained by the independent variables in a regression model. It is a crucial indicator of model fit, with a score of 1 signifying perfect prediction accuracy.
(13)
where:
- SSR (Sum of Squares of Residuals) calculates the sum of the squared differences between the observed values and the values predicted by the model. It is mathematically represented as:
, where are the actual values and denotes the prediction values.
- TSS (Total Sum of Squares) represents the overall variance within the dataset, determined by summing the squared deviations of each observed value from the dataset's mean:
, where represents the observed value mean.
- Some parameters are computed, but what they reveal about the body is unclear.
Response: Thank reviewer for your comments and suggestion. We add a table to describe these gait parameters more clear in Table 1
Table 1. Description of 17 temporospatial gait parameters.
|
Type |
Name |
Units |
Description |
|
Temporal Parameter
|
Heel-Strike (HS) |
Seconds |
The moment the heel makes contact with the ground. |
|
Gait Cycle Time (GCT) |
Steps/ minute |
Gait cycle duration is the time between heel strikes on the same foot. |
|
|
Double Leg Support (DS) |
% of cycle duration |
The bipedal stance period is when both feet are on the ground during the gait cycle. |
|
|
Cadence |
Steps/ minute |
The number of steps walked per minute. |
|
|
Stance |
% of cycle duration |
The foot hits the ground in the gait cycle's stance phase. |
|
|
Swing |
% of gait cycle |
The swing phase refers to the interval when the foot is not in contact with the ground during the gait cycle. |
|
|
Foot Flat Ratio (FFr) |
% of stance |
The phase of the stance in which the foot is completely in contact with the ground, with the sole entirely touching the surface. |
|
|
Push Ratio (Purify) |
% of stance |
The time between flat soles and lifted toes in stance. |
|
|
|
Load Ratio (LDr) |
% of stance |
The stance period from heel strike to full sole contact. |
|
Spatial Parameters |
Step Length |
meters |
The spatial distance between the feet when positioned on the ground. |
|
Stride Length |
meters |
Distance between heel strikes, equaling one gait cycle. |
|
|
Gait Speed |
meters/s |
Speed of forward walking. |
|
|
Peak Swing |
meters/s |
Maximum angular velocity from heel to toe during swing. |
|
|
Foot Pitch Angle at Heel Strike (HSP) |
degree |
The angle formed by the foot's contact with the ground upon impact. |
|
|
Foot Pitch Angle at Toe-Off (TOP) |
degree |
The angle of the toes relative to the ground just before lift-off at the end of the propulsion phase. |
|
|
Swing Width |
meters |
The largest sideways distance in the swing phase corresponds to the maximum lateral offset. |
|
|
3D Path Length |
% of stride length |
Depicts the scaled trajectory of the 3D gait cycle using stride length. |
- “development of a ML model to predict gait parameters” - which is the developed algorithm? As I understand you use an algorithm already available. Please detail what means “the development”!
Response: Thank reviewer for your comments and suggestion. We change the word ‘development’ to ‘use’.
Line: 431-444
- Conclusions
This study marks a notable advancement in biomechanics, specifically in the analysis of leg movements, by integrating gait analysis with sEMG and ACC data. Utilizing advanced wireless technology, researchers gathered a dataset from 17 participants, enhancing our understanding of human locomotion. A significant aspect of this research is the use of a ML model to predict gait parameters from sEMG signals of thigh and calf muscles, along with ACC signals, demonstrating high accuracy. The CatBoost model's superiority over other models, such as XGBoost and Decision Tree, especially in processing complex high-frequency sEMG and ACC signals, is a key finding. This suggests potential for broader application in biomechanical data analysis. The study has limitations, including small datasets and challenges in improving the Pearson Correlation Coefficient (PCC) results, as it could not accurately predict Heel Strike (HS) and Path Length. Future studies should consider enlarging the sample population and integrating neural network and deep learning approaches to achieve a more thorough and dependable analysis of gait.
- The introduction must be improved. You have to offer more details about sEMG working principle, so the reader can understand how the signals are connected to the muscle activities.
Response: Thank reviewer for your comments and suggestion. We extend the Introduction section of our paper. Lines 27 -107
- Introduction
The gait, defined as the distinctive movement pattern of the lower extremities during ambulation, serves as a key indicator of the human body's locomotive attributes [1]. This biomechanical phenomenon is integral to the control mechanisms of prosthetic legs for lower limb amputees. Additionally, gait analysis plays a critical part in the realms of individual identification, fall risk assessment, and the diagnostic process for various disorders, including Parkinson's disease [2][3]. Traditional approaches to leg movement analysis, predominantly reliant on observational techniques and complex motion capture systems, provide substantial insights but are often constrained by practical, cost, and ecological validity factors.
Surface electromyography (sEMG) is instrumental in capturing the electrical activity of muscles during leg movement, providing invaluable data on muscle coordination and activation patterns [3]. In exploring the detailed interplay between neuromuscular system and physical movement, sEMGs play a crucial role [4]. They yield vital insights into the patterns of muscle activation, the robustness of muscular strength, and the dynamics of muscle exhaustion [5]. sEMG, an instrument essential for capturing the electrical fields emitted by muscle fibers upon contraction, utilizes motor unit action potentials (MUAPs) as its primary signal. These MUAPs, which represent the electrical patterns of muscle fibers, may increase in intensity or frequency based on the muscle's activity level. The sEMG methodology includes a system that amplifies and visually represents these electrical signals, thereby enabling researchers and clinicians to monitor muscle activation, timing, and coordination [6]. This capability is critically important for assessing muscle function, diagnosing neuromuscular disorders such as Myopathy, managing strength training, providing biomechanical insights, assisting in the design of orthotics and prosthetics, and advancing the development of ergonomic solutions [7]. sEMG in gait analysis proves beneficial for patients with neuromuscular conditions such as cerebral palsy, Parkinson's disease and muscle dystrophy [8]. Several researches have employed accelerometer (ACC) for the identification of gait parameters [9,10].
Recent advancements in gait analysis have focused on integrating inertial measurement unit (IMU) and sEMG data to enhance the estimation of gait parameters. These methods employ both types of sensors to capture comprehensive biomechanical movements, combining the kinematic data from IMUs with the kinetic data from sEMG sensors for a holistic view of gait dynamics. Notably, machine learning (ML) models namely, Random Forests (RF), Neural Networks (NNs), and Support Vector Machines (SVM) are increasingly utilized to analyze the rich, multidimensional data sets provided by these sensors. This approach not only improves the accuracy of gait parameter estimations but also facilitates real-time gait analysis, making it invaluable in clinical and sports settings for immediate feedback and continuous monitoring [11,12].
The use of ML methods facilitates adaptive learning from extensive datasets, which can enhance the accuracy and effectiveness of diagnostics and monitoring in clinical environments [13]. ML has demonstrated its efficiency in analyzing sEMG signals for various applications, including gesture classification [14], muscle fatigue detection [15][16], and identify sarcopenia [2]. In the previous research Kidziński et al. [17] employed a data-driven method to forecast the timings of foot-contact and foot-off events by analyzing marker time series and kinematic in children exhibiting both normal and abnormal walking patterns. Arunganesh et al. [18] applied a tree-based ML model to identify lower limb movements using sEMG measurements. In another study, Rastegari et al. [19] and Trabassi et al. [20] used ML models to identify Parkinson's disease through gait analysis based on ACC data. Howcroft et al. [21] utilized different ML techniques to estimate the risk of falls using ACC parameters. Zhang et al. [22] used Support Vector Regression (SVR) models to accurately estimate essential gait phases. Jani et al. [23] employed ML techniques such as CatBoost, Random Forest (RF), XGBoost, LightGBM, and Decision Tree (DT) for detecting gait abnormalities. The outcome of this study indicated that CatBoost achieved the highest accuracy compared to the other ML techniques. Wu et al. [24] used traditional ML such as XGBoost, RF, and Linear Discriminant Analysis (LDA) for the categorization of lower limb movements. Among these, XGBoost performed the best. Among these, XGBoost performed well. Armand et al. [25] used fuzzy decision trees to link clinical data with kinematic gait patterns of toe-walking. In the existing literature, only Liu et al. [26] have predicted 11 temporospatial gait parameters, whereas other studies [27] [22] have typically predicted between 2 and 6 gait parameters.
This study focuses on examining the temporal and spatial parameters of walking patterns using sEMG and ACC data of lower limbs, analyzed through ML techniques such as CatBoost, XGBoost and DT. This study selected CatBoost, XGBoost, and DT due to their strong performance in handling complex datasets, as similar studies [23][24][25]. CatBoost efficiently manages categorical data, while XGBoost is valued for its anti-overfitting capabilities. DT were chosen for their clear interpretability, which is essential in clinical applications. Together, these classifiers offer a balanced approach to our analysis. The sEMG data were collected from two primary muscles in each foot, the vastus lateralis and gastrocnemius muscles. ACC were placed at the thighs. For benchmarking, the study utilized the GaitUp Physilog®5 shoe-worn inertial sensors, a professional gait analysis tool [28]. A wireless self-made multi-channel measurement system was employed to record muscle activity and thigh motion during treadmill running. The main contribution of this study:
- Analyzed ACC and sEMG signals to extract features relevant to gait analysis, enhancing the understanding of gait dynamics.
- Applied feature selection methods to recognize the important features that contribute to the accuracy of the gait parameter predictions.
- Employed ML techniques to predict temporal and spatial 17 gait parameters.
Following this introduction, the study organizes the paper into several key sections. The next section details the materials and methods employed in our study. The results and discussion section follows, providing an in-depth interpretation of the findings. The study concludes with a summary of these findings.
- Give more details about the used algorithms Catboost, XGBoost, and DT.
Response: Thank reviewer for your comments and suggestion. We add more details in section 2.5. Line:254- 285
2.5.1. CatBoost
CatBoost [36] is an advanced open-source ML algorithm developed by Yandex, renowned for its efficient handling of categorical data and high performance. It stands out in the gradient boosting landscape due to its native processing of categorical variables, eliminating the need for extensive preprocessing. CatBoost [37] utilizes a combination of categorical features, leveraging the relationships among these features to significantly enhance the dimensionality of the feature space. To minimize overfitting and enhance both the accuracy and generalizability of the model, CatBoost employs a perfectly symmetrical tree structure. Known for its speed and accuracy, CatBoost delivers robust results even with default settings, making it a favored choice among data scientists and researchers for a wide range of applications.
2.5.2 XGBoost
XGBoost, or eXtreme Gradient Boosting [38], represents an enhancement over traditional Gradient Boosting methods. Structurally similar to DT, it combines several weak DTs to create a powerful predictive tool. XGBoost generally outperforms standard classification and regression techniques in terms of accuracy. Due to its robust adaptive learning capabilities, XGBoost remains a favored choice for both regression and classification tasks in current study and competitive environments. XGBoost [39] addresses the issue of overfitting by regulating tree complexity and incorporating a regularization term into the objective function, enhancing model reliability.
2.5.3. Decision Tree
Decision Tree [40] is a model defined by a tree-like structure of hierarchy. Each node in this structure represents different decisions, ultimately guiding towards expected results. The unique setup of DTs makes them exceptionally transparent, allowing for straightforward understanding of the model's decision-making rules. As a result, DTs [41] have gained popularity as one of the leading nonlinear regression models. It’s especially skilled in handling nonlinear regression challenges, effectively capturing complex relationships between inputs and outputs without relying on linear associations. DT is capable of developing several linear regression models within the leaf nodes of the tree. Furthermore, DT visually demonstrate the significance of various contributors, thereby enhancing analytical support. Nonetheless, their dependency on categorical variables poses a significant challenge [42].
- Improve the Conclusions. May be a discussion in respect with other methods presented in the literature would help.
Response: Thank reviewer for your comments and suggestion. We add some sentences to compare with the previous studies in section 4, and modify the mentions of conclusion and the limitations of this study in section 5.
Line 416-430.
This study also compares our findings with previous studies. Zhang et al. [22] used ML to estimate three gait parameters—stride length, velocity, and foot clearance—with MAEs of 2.37 ± 0.65, 3.10 ± 0.57, and 0.58 ± 0.28, respectively, for running. The MAE for 17 gait parameters in this study, predicted by CatBoost, is 1.00, which represents an improvement over the previous study. Moreover, Liu et al. only estimate 11 gait parameters (six temporal parameters and five spatial parameters), PCC above 0.800, by sEMG signals [26]. In this study, the number of predicted gait parameters with PCC above 0.800 was fifteen when sEMG signals combining with ACC signals. In the another study, Wei et al. [27] utilized sEMG signals to recognize gait phases and predict lower limb kinematics. The research employed LDA, SVM, and LSTM as predictive models, collecting sEMG data from nine leg muscles to estimate four gait parameters: initial contact, foot flat, heel off, and toe off. The study found that LSTM achieved the highest performance with an accuracy of 93% and an RMSE of 1.86. In comparison, our study achieved a better RMSE of 1.46 and demonstrated superior PCC values. The results show that the IMU, ACC, could describe more detail leg-motion parameters when user walking or running.
Line 433-446.
- Conclusions
This study marks a notable advancement in biomechanics, specifically in the analysis of leg movements, by integrating gait analysis with sEMG and ACC data. Utilizing advanced wireless technology, researchers gathered a dataset from 17 participants, enhancing our understanding of human locomotion. A significant aspect of this research is the use of a ML model to predict gait parameters from sEMG signals of thigh and calf muscles, along with ACC signals, demonstrating high accuracy. The CatBoost model's superiority over other models, such as XGBoost and Decision Tree, especially in processing complex high-frequency sEMG and ACC signals, is a key finding. This suggests potential for broader application in biomechanical data analysis. The study has limitations, including small datasets and challenges in improving the PCC results, as it could not accurately predict HS and Path Length. Future studies should consider enlarging the sample population and integrating neural network and deep learning approaches to achieve a more thorough and dependable analysis of gait.
- Detail the utility of these parameters you estimate.
Response: Thank reviewer for your comments and suggestion. We add some sentences to mention the utility of these predicting gait parameters in section 4. Line 412-414.
These gait parameters are crucial for diagnosing gait disorders and designing assistive devices as well as the customization of prosthetics and orthotics.

Round 2
Reviewer 1 Report
Comments and Suggestions for Authors
Revised manuscript can be accepted after professional english editing of the manuscript.
Comments on the Quality of English LanguageEnglish editing is required
Author Response
To Reviewer #1:
Thank the reviewer for his/her valuable comments that make better this manuscript. The texts in this revised manuscript have been corrected/ modified by red words. It is our sincere hope that this revision will enhance readability and strengthen of the manuscript to satisfy the requirements of this prestigious journal.
Comments and Suggestions for Authors
Revised manuscript can be accepted after professional english editing of the manuscript
Response: Many thanks for reviewer's comment. This manuscript have been carefully edited English by third author again.
Reviewer 3 Report
Comments and Suggestions for Authors
-
remark 6: Who is Z’? How do you determine m and v? Your answer and added text are not clear.
-
remark 7 - you switch from y to a and b. You confuse the reader. The sums also do not have limits for the index
Author Response
To Reviewer #3:
Thank the reviewer for his/her valuable comments that make better this manuscript. The texts in this revised manuscript have been corrected/ modified by red words. It is our sincere hope that this revision will enhance readability and strengthen of the manuscript to satisfy the requirements of this prestigious journal.
Comments and Suggestions for Authors
- remark 6: Who is Z’? How do you determine m and v? Your answer and added text are not clear.
Response: Many thanks for reviewer’s comment, For Z’, “ , “ is equation “comma”. We also modify the mention of m and v values. Line 209-227.
Sample Entropy (SampEn) is used to calculate the intricacy and regularity of a signal's temporal sequence. An increased entropy value indicates time series complexity. When the length of a signal is S, calculating sample entropy requires defining dimension m and selecting a suitable v value. After determining m, segment signal labeled Xm. The method is outlined as stated below,
. (5)
Equation (5) is the formula for SampEn.
, (6)
where,
, (7)
. (8)
The function measures the distance between vectors and in the reconstructed phase space, where each vector represents a subsequence of length starting at points i and j, respectively. The limits for i and j in Equations (7) and (8) are intended to span the entire dimension of the time series, ensuring that all possible pairs of subsequences of length m and m+1 are considered in computation. Typically, i and j range from 1 to S-m or S-m-1. This research sets m=2 and v = STD × 0.2. The variables m and v in our SampEn calculations are chosen based on our best adjustment, and following the methodology of previous studies [31][32].
- remark 7 - you switch from y to a and b. You confuse the reader. The sums also do not have limits for the index
Response: We modify the mention of PCC. Line 317-324.
This study presents the numerical data as a mean (M) ± standard deviation (STD). To elucidate the connection between the target and predicted values observed in our test dataset, we have employed the Pearson correlation coefficient (PCC). The mathematical formulation of this coefficient is explicitly detailed in Equation (14),
, (14)
where and corresponds to the mean values of the actual values, Yi, and the prediction values, , respectively. This statistical approach enables a rigorous evaluation of the linear correlation between the two sets of variables under investigation.
